# Deep Fourier Up-Sampling

**Man Zhou**[1,2]* **Hu Yu**[2]* **Jie Huang**[2], **Feng Zhao**[2],
**Jinwei Gu**[6], **Chen Change Loy**[3], **Deyu Meng**[4,5], **Chongyi Li**[3]†
[1]Hefei Institute of Physical Science, Chinese Academy of Sciences, China
[2]University of Science and Technology of China, China
[3]S-Lab, Nanyang Technological University, Singapore
[4]Xi'an Jiaotong University, China
[5]Pazhou Laboratory (Huangpu), China
[6]SenseBrain Technology (SenseTime Research USA), USA
{manman,yuhu520,hj0117}@mail.ustc.edu.cn, fzhao956@ustc.edu.cn,
gujinwei@sensebrain.ai, dymeng@mail.xjtu.edu.cn
{ccloy,chongyi.li}@ntu.edu.sg
https://li-chongyi.github.io/FourierUp_files/

## Abstract

Existing convolutional neural networks widely adopt spatial down-/up-sampling for multi-scale modeling. However, spatial up-sampling operators (*e.g.*, interpolation, transposed convolution, and un-pooling) heavily depend on local pixel attention, incapably exploring the global dependency. In contrast, the Fourier domain obeys the nature of global modeling according to the spectral convolution theorem. Unlike the spatial domain that performs up-sampling with the property of local similarity, up-sampling in the Fourier domain is more challenging as it does not follow such a local property. In this study, we propose a theoretically sound Deep Fourier Up-Sampling (FourierUp) to solve these issues. We revisit the relationships between spatial and Fourier domains and reveal the transform rules on the features of different resolutions in the Fourier domain, which provide key insights for FourierUp's designs. FourierUp as a generic operator consists of three key components: 2D discrete Fourier transform, Fourier dimension increase rules, and 2D inverse Fourier transform, which can be directly integrated with existing networks. Extensive experiments across multiple computer vision tasks, including object detection, image segmentation, image de-raining, image dehazing, and guided image super-resolution, demonstrate the consistent performance gains obtained by introducing our FourierUp. Code is available at `https://manman1995.github.io/`.

## 1 Introduction

Spatial down-/up-sampling has been widely used in convolutional neural networks for multi-scale modeling. For example, U-Net [1], a variation of encoder-decoder, employs pooling layers to reduce the feature resolution in the encoder and then recovers the resolution using up-sampling operations in the decoder. In addition, the feature pyramid [2–5] and image pyramid [6–9] driven multi-scale neural networks rely on the down-/up-sampling operation to obtain multi-scale property and improve modeling capability. However, spatial up-sampling operators (*e.g.*, interpolation, transposed convolution, and un-pooling) heavily depend on local pixel attention, and thus cannot

---

*Man Zhou and Hu Yu contribute equally.
†Corresponding author: Chongyi Li.

36th Conference on Neural Information Processing Systems (NeurIPS 2022).

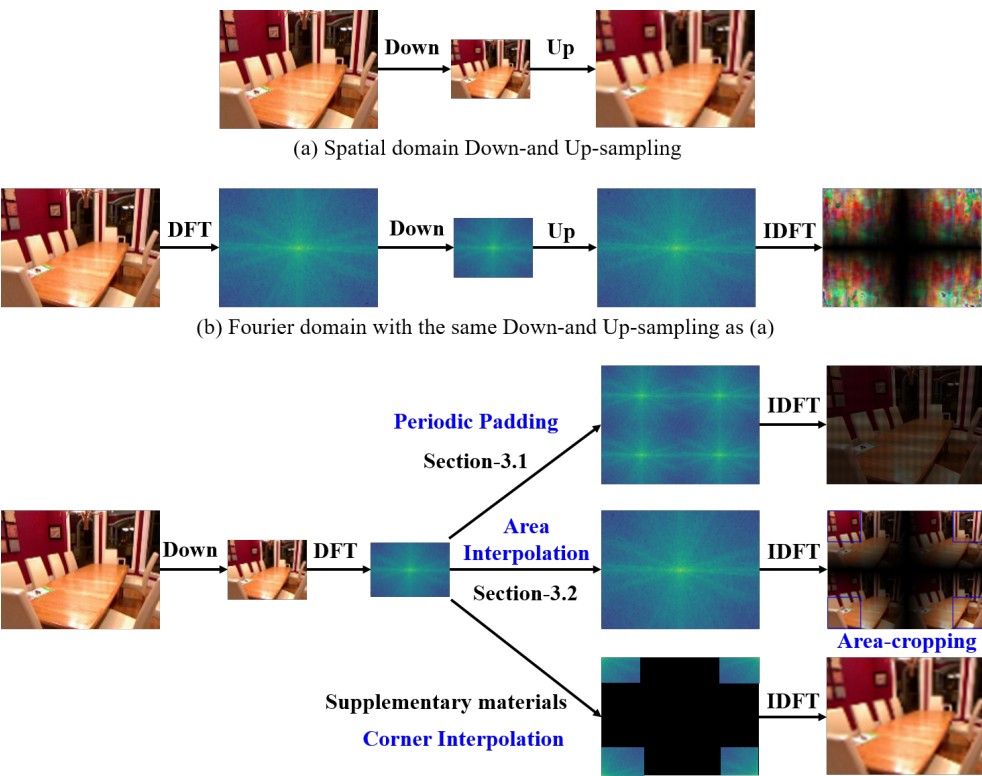

(a) Spatial domain Down-and Up-sampling

(b) Fourier domain with the same Down-and Up-sampling as (a)

(c) Fourier domain with our proposed "**Fourier Up-sampling**"

Figure 1: **Motivation.** (a) and (b) depict that arbitrary up-sampling, *e.g.*, interpolation, in the Fourier domain produces sub-optimal result as it does not follow the same local similarity property as that in the spatial domain. This motivates us to design a more ingenious "Fourier Up-Sampling" operator, dubbed as FourierUp. It has three alternative variants: Periodic Padding, Area Interpolation/Cropping and Corner Interpolation, as illustrated in (c).

explore the global dependency that is indispensable for many computer vision tasks [10, 11, 1, 12–20]. According to the spectral convolution theorem, the Fourier domain obeys the nature of global modeling, providing an alternative solution for multi-scale modeling. However, unlike the spatial domain with local similarity property, up-sampling in the Fourier domain is more challenging as it does not follow such a local property. The observation encourages us to explore deep Fourier up-sampling.

Recent studies have explored information interaction in both spatial and Fourier domains. FFC [21], for instance, replaces the conventional convolution with a spatial-Fourier interaction, which consists of a spatial (or local) path that performs conventional convolution on a portion of input feature channels and a spectral (or global) path that operates in the Fourier domain. DFT [22] devises a Residual Fast Fourier Transform Block to integrate both low- and high-frequency residual information by performing the interaction between a regular spatial residual stream and a channel-wise Fourier transform stream. However, the aforementioned methodologies only interact at a single resolution scale, and the spatial-Fourier interaction potential of multiple scales in the Fourier domain has not been investigated. The key to solving this problem lies in how to implement deep Fourier up-sampling for multi-scale Fourier pattern modeling.

**Challenges.** Owing to the local similarity and cross-scale position invariant properties of the spatial domain, the various spatial up-sampling operations including transposed convolution, un-pooling, and interpolation techniques are capable of using the pixel neighboring relationship to interpolate the unknown pixel values at local regions, increasing the spatial resolution of the features, as shown in Figure 1(a). In contrast to the spatial domain, the Fourier domain does not share the same scale-invariant property and local texture similarity, and hence cannot implement up-sampling using the same techniques as the spatial domain, as illustrated in Figure 1(b).

**Solutions.** In this paper, we wish to investigate the possibility of devising a reliable up-sampling in the Fourier domain in a theoretical sound manner. To answer this question, we first revisit the relationship between spatial and Fourier domains, revealing the transform rules on the features of different resolutions in the Fourier domain (see Section 3.1 and Section 3.2). On the basis of the above rules, we propose a theoretically feasible Deep Fourier Up-Sampling (FourierUp). Specifically, we develop three variants (Periodic Padding, Area Interpolation/Cropping and Corner Interpolation) of FourierUp (see Section 3.3), as illustrated in Figure 1(c). Each variant consists of three key components: 2D discrete Fourier transform, Fourier dimension increase rules, and 2D inverse Fourier transform. FourierUp is a generic operator that can be directly integrated with existing networks. Extensive experiments on multiple computer vision tasks, including object detection, image segmentation, image de-raining, image dehazing, and guided image super-resolution, demonstrate the consistent performance gains obtained by introducing our FourierUp. *We believe that the proposed FourierUp could refresh the neural network designs where the spatial and Fourier information interaction at only a single resolution scale are mainstream choices.*

**Contributions.** 1) We propose Deep Fourier Up-Sampling, a novel method that enables the integration of the features of different resolutions in the Fourier domain. This is the first thorough effort to explore the Fourier up-sampling for multi-scale modeling. 2) The proposed FourierUp is a generic operator that can be directly integrated with the existing networks in a plug-and-play manner. 3) Equipped with the theoretically sound FourierUp, we show that existing networks could achieve consistent performance improvement across multiple computer vision tasks.

## 2 Related Work

**Spatial Up-Sampling.** Convolutional neural networks with spatial down-/up-sampling have become the *de facto* structures in many computer vision tasks [23–31]. Typically, U-Net [1] builds multi-scale feature maps using the encoder with down-sampling and then utilizes the up-sampling operation to fuse the multi-scale features in the decoder. Additionally, the feature pyramid [2–5] and image pyramid[6–9] are commonly used to obtain the multi-scale property in neural networks [6–9]. Among them, spatial up-sampling plays a significant role in multi-scale modeling. However, existing up-sampling operations only work in the spatial domain and current studies rarely explore the potential (*e.g.*, the global modeling capability) of up-sampling in the frequency domain.

**Spatial-Fourier Interaction.** Recently, several studies attempt to employ Fourier transform in deep models [32–35, 21]. Some of these efforts use discrete Fourier transform to transfer the spatial features to the Fourier domain and then use frequency information to improve the performance of particular tasks [32, 34]. Another line is to use convolution theorem to speed up the models, such as using fast Fourier transform (FFT) [35, 21]. For example, FFC [21] replaces the convolution with the spatial-Fourier interaction. The work proposed in [36] uses spectral pooling to reduce feature resolution by truncating the frequency domain representation. However, all the techniques only interact with each other at a single spatial resolution and have not explored the interaction potential at multiple resolutions in both spatial and frequency domains as performing the frequency up-sampling is non-trivial. As a tentative exploration, we study the relationship between the spatial domain and Fourier domain and reveal the transform rules over the feature of different resolutions in the Fourier domain. This delivers the underlying insights for the designs of multi-scale Fourier modeling patterns, which has the potential of versatility for different network architectures.

## 3 Deep Fourier Up-Sampling

We first explore the mapping relationship between the spatial and Fourier domains, and then present three Deep Fourier up-sampling variants, including i) periodic padding of magnitude and phase, ii) area up-sampling of magnitude and phase, and iii) corner interpolation of magnitude and phase, based on the explored transform rules. In terms of the first two variants, we provide two theorems and their proofs as follows while the third is reported in supplementary materials.

**Definitions.** $f(x, y) \in \mathbb{R}^{2M \times 2N}$ is the 2-times zero-inserted up-sampled version of $g(x, y) \in \mathbb{R}^{M \times N}$ in spatial domain, and $F(u, v) \in \mathbb{R}^{2M \times 2N}$, $G(u, v) \in \mathbb{R}^{M \times N}$ denote their Fourier transforms. $H(u, v) \in \mathbb{R}^{2M \times 2N}$ is the 2-times area-interpolation up-sampled Fourier transform of $G(u, v)$, and $h(x, y) \in \mathbb{R}^{M \times N}$ denotes their inverse Fourier transform.

**Theorem-1.** $F(u, v) = F(u + M, v) = F(u, v + N) = F(u + M, v + N)$ and $G(u, v) = \frac{F(u,v)}{4}$ where $u = 0, 1, 2, \ldots, N - 1$ and $v = 0, 1, 2, \ldots, M - 1$. $F(u, v)$ is exactly the periodic padding of $G(u, v)$ where $G(u, v)$ is exactly the quarter of $F(u, v)$ with the value being $\frac{1}{4}$ times decay.

**Theorem-2.** $H(2u, 2v) = H(2u + 1, 2v) = H(2u, 2v + 1) = H(2u + 1, 2v + 1) = G(u, v)$ with $u = 0, 1, \ldots, M - 1$ and $v = 0, 1, \ldots, N - 1$ and

$$
\begin{aligned}
h(x, y) &= \frac{A(x, y)}{4} g(x, y) \\
h(x + M, y) &= \frac{A(2M - x, y)}{4} g(x, y) \\
h(x, y + N) &= \frac{A(x, 2N - y)}{4} g(x, y) \\
h(x + M, y + N) &= \frac{A(2M - x, 2N - y)}{4} g(x, y)
\end{aligned}
\tag{1}
$$

where $A(x, y) = 1 + e^{\frac{j\pi x}{M}} + e^{\frac{j\pi y}{N}} + e^{j\pi(\frac{x}{M} + \frac{y}{N})}$ and $x = 0, 1, \ldots M - 1, y = 0, 1, \ldots N - 1$.

**Theorem-3.** Suppose the corner interpolated $F_G^{cor}(u, v)$ of the Fourier map $G(u, v) \in \mathbb{R}^{M \times N}$, it holds that the inverse Fourier transform $f_g^{cor}(x, y)$ of $F_G^{cor}(u, v)$

$$
f_g^{cor}(x, y) = g(\frac{x'}{2}, \frac{y'}{2}) e^{j\pi(\frac{x'}{2} + \frac{y'}{2})} \frac{(-1)^{(x+y)}}{4},
\tag{2}
$$

where $x' = 2x$ and $y' = 2y$, $x = 0, 1, \ldots, M - 1$ and $y = 0, 1, \ldots, N - 1$.

### 3.1 Proof-1 of Theorem-1: Periodic Padding of Magnitude and Phase

Note that $f(x, y) \in \mathbb{R}^{2M \times 2N}$ is up-sampled over $g(x, y) \in \mathbb{R}^{M \times N}$ by a factor of 2. The relationship between $g(x, y)$ and $f(x, y)$ can be written as

$$
\boldsymbol{f}(x, y) = \begin{cases} g(\frac{x}{2}, \frac{y}{2}), & x = 2m, y = 2n \\ 0, & \text{others} \end{cases}
\tag{3}
$$

where $m = 1, 2, \ldots, M - 1$ and $n = 1, 2, \ldots, N - 1$, the Fourier transform $F(u, v)$ of $f(x, y)$ is expressed as

$$
\begin{aligned}
F(u, v) &= \frac{1}{4MN} \sum_{x=0}^{2M-1} \sum_{y=0}^{2N-1} f(x, y) e^{-j2\pi(\frac{ux}{2M} + \frac{vy}{2N})} \\
&= \frac{1}{4MN} \sum_{x=0}^{M-1} \sum_{y=0}^{N-1} f(2x, 2y) e^{-j2\pi(\frac{u(2x)}{2M} + \frac{v(2y)}{2N})} \\
&= \frac{1}{4MN} \sum_{x=0}^{M-1} \sum_{y=0}^{N-1} f(2x, 2y) e^{-j2\pi(\frac{ux}{M} + \frac{vy}{N})} \\
&= \frac{1}{4MN} \sum_{x=0}^{M-1} \sum_{y=0}^{N-1} g(x, y) e^{-j2\pi(\frac{ux}{M} + \frac{vy}{N})}.
\end{aligned}
\tag{4}
$$

Then, we show the periodicity of $F(u, v) \in R^{2M \times 2N}$ with $M$ and $N$. It means $F(u, v) = F(u + M, v) = F(u, v + N) = F(u + M, v + N)$ with $u = 0, 1, 2, \ldots, N - 1$ and $v = 0, 1, 2, \ldots, M - 1$. We take the $F(u, v) = F(u + M, v)$ for example and recall Eq. (4) as

$$
\begin{aligned}
F(u + M, v) &= \frac{1}{4MN} \sum_{x=0}^{M-1} \sum_{y=0}^{N-1} f(2x, 2y) e^{-j2\pi(\frac{(u+M)x}{M} + \frac{vy}{N})} \\
&= \frac{1}{4MN} \sum_{x=0}^{M-1} \sum_{y=0}^{N-1} f(2x, 2y) e^{-j2\pi(\frac{ux}{M} + \frac{vy}{N})} e^{-2j\pi x} \\
&= \frac{1}{4MN} \sum_{x=0}^{M-1} \sum_{y=0}^{N-1} f(2x, 2y) e^{-j2\pi(\frac{ux}{M} + \frac{vy}{N})} \\
&= F(u, v),
\end{aligned}
\tag{5}
$$

where $e^{-2\pi jx} = 1$ for any integer $x$. Similarly, we can proof the periodicity of $F(u,v)$ as well.

Based on the above proof, the DFT of $g(x,y)$ can be formulated as:

$$G(u,v) = \frac{1}{MN} \sum_{x=0}^{M-1} \sum_{y=0}^{N-1} g(x,y)e^{-j2\pi(\frac{ux}{M}+\frac{vy}{N})}. \tag{6}$$

Revising Eq. (4), we can figure out that $G(u,v) = \frac{F(u,v)}{4}$.

## 3.2 Proof-2 of Theorem-2: Area Interpolation of Magnitude and Phase

The 2D Inverse Discrete Fourier transform (IDFT) of $G(u,v)$ can be written as:

$$g(x,y) = \frac{1}{MN} \sum_{u=0}^{M-1} \sum_{v=0}^{N-1} G(u,v)e^{j2\pi(\frac{ux}{M}+\frac{vy}{N})}. \tag{7}$$

We up-sample $G(u,v)$ with a size of $M \times N$ to get $H(u,v)$ with a size of $2M \times 2N$. Specifically, the area interpolation shown in Figure 3(b) is used for interpolation and then the interpolated pixels are the same as the original pixel in the $2 \times 2$ local regions. Namely, $H(2u,2v) = H(2u+1,2v) = H(2u,2v+1) = H(2u+1,2v+1) = G(u,v)$ with $u = 0,1,\ldots,M-1$ and $v = 0,1,\ldots,N-1$. Similar to Eq. (4), we can infer

$$
\begin{aligned}
h(x,y) &= \frac{1}{4MN} \sum_{u=0}^{2M-1} \sum_{v=0}^{2N-1} H(u,v)e^{j2\pi(\frac{ux}{2M}+\frac{vy}{2N})} \\
&= \frac{1}{4MN} \sum_{u=0}^{M-1} \sum_{v=0}^{N-1} H(2u,2v)e^{j2\pi(\frac{2ux}{2M}+\frac{2vy}{2N})} + \frac{1}{4MN} \sum_{u=0}^{M-1} \sum_{v=0}^{N-1} H(2u+1,2v)e^{j2\pi(\frac{(2u+1)x}{2M}+\frac{2vy}{2N})} \\
&\quad + \frac{1}{4MN} \sum_{u=0}^{M-1} \sum_{v=0}^{N-1} H(2u,2v+1)e^{j2\pi(\frac{2ux}{2M}+\frac{(2v+1)y}{2N})} + \frac{1}{4MN} \sum_{u=0}^{M-1} \sum_{v=0}^{N-1} H(2u+1,2v+1)e^{j2\pi(\frac{(2u+1)x}{2M}+\frac{(2v+1)y}{2N})} \\
&= \frac{1}{4MN} \sum_{u=0}^{M-1} \sum_{v=0}^{N-1} G(u,v)e^{j2\pi(\frac{ux}{M}+\frac{vy}{N})}[1 + e^{\frac{j\pi x}{M}} + e^{\frac{j\pi y}{N}} + e^{j\pi(\frac{x}{M}+\frac{y}{N})}].
\end{aligned}
\tag{8}
$$

Similarly, we can write $g(x,y)$ as

$$g(x,y) = \frac{1}{MN} \sum_{u=0}^{M-1} \sum_{v=0}^{N-1} G(u,v)e^{j2\pi(\frac{ux}{M}+\frac{vy}{N})}. \tag{9}$$

Recalling Eq. (8) and Eq. (10), we can infer

$$h(x,y) = \frac{1 + e^{\frac{j\pi x}{M}} + e^{\frac{j\pi y}{N}} + e^{j\pi(\frac{x}{M}+\frac{y}{N})}}{4} g(x,y). \tag{10}$$

where $x = 0,1,\ldots M-1, y = 0,1,\ldots N-1$. We remark $1 + e^{\frac{j\pi x}{M}} + e^{\frac{j\pi y}{N}} + e^{j\pi(\frac{x}{M}+\frac{y}{N})} = A(x,y)$, $|A(x,y)|^2 = (1 + \cos\pi\frac{x}{M} + \cos\pi\frac{y}{N} + \cos\pi(\frac{x}{M}+\frac{y}{N}))^2 + (\sin\pi\frac{x}{M} + \sin\pi\frac{y}{N} + \sin\pi(\frac{x}{M}+\frac{y}{N}))^2$.

We can find that the variable $x$ shares the same operations as the variable $y$. For brevity, we only take the operation of variable $x$ as example

$$
\begin{aligned}
\frac{\partial |A(x,y)|^2}{\partial x} &= \frac{2\pi}{M}(1 + \cos\pi\frac{x}{M} + \cos\pi\frac{y}{N} + \cos\pi(\frac{x}{M}+\frac{y}{N}))(-\sin\pi\frac{x}{M} - \sin\pi(\frac{x}{M}+\frac{y}{N})) \\
&\quad + \frac{2\pi}{M}(\sin\pi\frac{x}{M} + \sin\pi\frac{y}{N} + \sin\pi(\frac{x}{M}+\frac{y}{N}))(\cos\pi(\frac{x}{M}) + \cos\pi(\frac{x}{M}+\frac{y}{N})) \\
&= -\frac{4\pi}{M}(\sin\pi\frac{x}{M})(1 + \cos\pi(\frac{y}{N})).
\end{aligned}
\tag{11}
$$

Equally, we have

$$\frac{\partial |A(x,y)|^2}{\partial y} = -\frac{4\pi}{N}\sin\pi\frac{y}{N}(1 + \cos\pi(\frac{x}{M})). \tag{12}$$

We prove that the partial derivative of $|A(x,y)|$ on both x and y is negative for $x \in [0, M-1]$ and $y \in [0, N-1]$. Besides, we have

$$|A(x,y)| = |A(2M-x,y)| = |A(2M,2N-y)| = |A(2M-x,2N-y)|. \tag{13}$$

That is to say, the intensity drops from the side to the center, shown in Figure 4. Specifically, the intensity drops to zero at the position of $x = M$ or $y = N$.

```
def DFU_Padding(X):                         def DFU_AreaInterpolation(X):
 # X: input with shape [N, C, H, W]          # X: input with shape [N, C, H, W]
 # A and P are the amplitude and phase       # A and P are the amplitude and phase
    A, P = FFT(X)                               A, P = FFT(X)

    # Fourier up-sampling transform              # Fourier up-sampling transform rules
        rules                                   A_aip = Area-Interpolation(A)
    A_pep = Periodic-Padding(A)               P_aip = Area-Interpolation(P)
    P_pep = Periodic-Padding(P)               A_aip = Convs_1x1(A_aip)
    A_pep = Convs_1x1(A_pep)                  P_aip = Convs_1x1(P_aip)
    P_pep = Convs_1x1(P_pep)
                                                # Inverse Fourier transform
    # Inverse Fourier transform               Y = iFFT(A_aip, P_aip)
    Y = iFFT(A_pep, P_pep)
                                                #Area Cropping
    Return Y #[N, C, 2H, 2W]                  Y = Area-Cropping(Y)
                                                Y = Resize(Y)

                                                Return Y #[N, C, 2H, 2W]
```

Figure 2: **Pseudo-code of the two variants of the proposed deep Fourier up-sampling.** The left is the *periodic padding variant* while the right is the *area interpolation-cropping variant*.

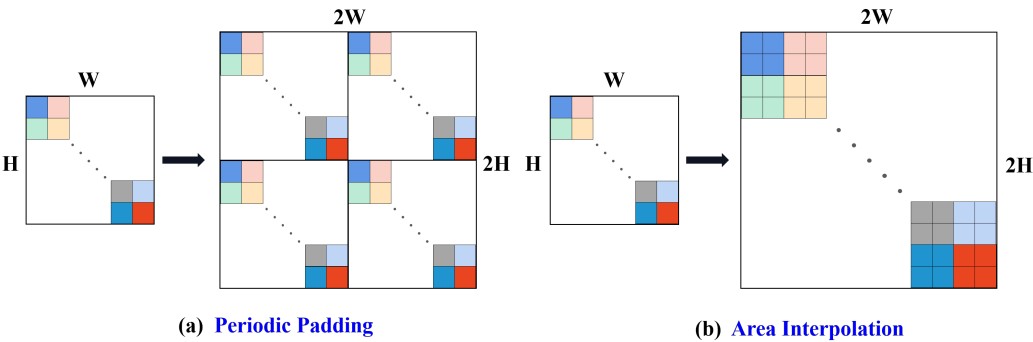

(a) **Periodic Padding**         (b) **Area Interpolation**

Figure 3: **The illustrations of (a) periodic padding and (b) area interpolation in Figure 2.** Each small color square represents a pixel of the amplitude/phase component in the Fourier domain.

## 3.3 Architectural Design

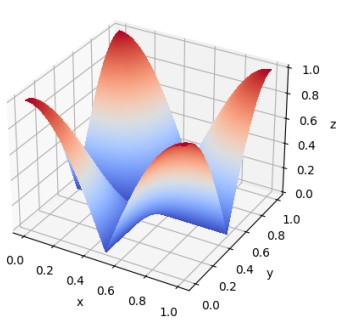

Figure 4: **The surface of $A(x, y)$.**

Recall the **Theorem-1** and **Theorem-2**, we propose two deep Fourier up-sampling variants: Periodic Padding and Area Interpolation-Cropping.

**Periodic Padding Up-Sampling.** The pseudo-code of Periodic Padding Up-Sampling is shown in the left of Figure 2. Given an image $X \in \mathbb{R}^{H \times W \times C}$, we first adopt the Fourier transform FFT(X) to obtain its amplitude component A and phase component P. We then perform the periodic padding over A and P two times in both the $H$ and $W$ dimensions, as illustrated in Figure 3(a). The padded A_pep and P_pep are then fed into two independent convolution module with $1 \times 1$ kernel and followed by the inverse Fourier transform iFFT(.) to project the padded ones back to spatial domain.

**Area Interpolation-Cropping Up-Sampling.** The pseudo-code of Area interpolation-Cropping Up-Sampling is shown in the right of Figure 2. We first conduct the Area interpolation over the phase and amplitude by $2 \times 2$ area interpolation with the same pixel, as illustrated in Figure 3(b). We then employ the inverse Fourier transform to project the interpolated ones back to spatial domain. As described in Section 3.2, the

inverse spatial representation will be periodic while the pixel value will be decay. The degree of decay of the pixel increases when the pixel is closer to the center. To better maintain the information, we perform the area cropping operation (detailed in Figure 1) in the four corners with the $\frac{W}{2} \times \frac{H}{2}$ size and then merge them together as a whole at spatial dimension, finally resize them to the size of $2H \times 2W$.

Note that albeit being designed on the basis of strict theories, both constructed spectral up-sampling modules contain certain approximations, like a learnable $1 \times 1$ convolution operator instead of strictly $1/4$ as described in Theorem-1 of main manuscript, and an approximation cropping to preserve the map corners instead of accurate $A$ mapping as proved in Theorem-2 of main manuscript and Theorem-3 of supplementary materials. Such strategy makes the proposed modules able to be more easily implemented and more flexibly represent real data spectral structures. It is worth noting that this should be the first attempt for constructing easy equitable spectral upsampling modules, and hope it would inspire more effective and rational ones from more spectral perspectives.

## 4  Experiments

To demonstrate the efficacy of our proposed deep Fourier up-sampling, we conduct extensive experiments on multiple computer vision tasks, including object detection, image segmentation, image de-raining, image dehazing, and guided image super-resolution. We provide more experimental results in the supplementary material.

### 4.1  Experimental Settings

**Object Detection.** Following [12], the PASCAL VOC 2007 and 2012 training sets [37] are used as training data. The PASCAL VOC 2007 testing set is used for evaluations as the ground truth annotations of VOC 2012 testing set are not publicly available. We employ the FPN-based Faster RCNN [12] with ResNet50 backbone and YOLO-v3 with Darknet53 [38] as baselines.

**Image Segmentation.** Following [39, 40], Synapse Dataset and CANDI Dataset are used as the testbed of medical image segmentation. We adopt the two representative image segmentation algorithms, U-Net [1] and Att-UNet [41], as the base models.

**Image De-raining.** Following [42], we choose two widely-used standard benchmark datasets, including Rain200H and Rain200L, for evaluations. we employ two representative de-raining methods, LPNet with up-sampling [11] and PReNet without up-sampling [42], as baselines.

**Image Dehazing.** Following [10], we employ RESIDE[43] dataset [44] for evaluations. We also use two different network designs AODNet [45] without up-sampling operator and MSBDN [10] with up-sampling operator, for validation.

**Guided Image Super-resolution.** Following [14, 46], we adopt the pan-sharpening, the representative task of guided image super-resolution, for evaluations. The WorldView II, WorldView III, and GaoFen2 in [14, 46] are used for evaluations. We employ two different network designs for validation, including PANNET [47] without up-sampling operator and DCFNET [48] with up-sampling operator.

Several widely-used image quality assessment (IQA) metrics are employed to evaluate the performance, including the relative dimensionless global error in synthesis (ERGAS) [49], the peak signal-to-noise ratio (PSNR), the spectral angle mapper (SAM) [50], DSC, and HD95.

### 4.2  Implementation Details

Regarding the above competitive baselines, they can be divided into two categories: one with spatial up-sampling ([1], Att-UNet [41], DCFNET [48], LPNet [11], MSBDN [10]) and another one without spatial up-sampling (PReNet [42], AODNet[45], PANNET [47]). The purpose of the exploration on the baselines without spatial up-sampling is to show the versatility of our FourierUp for different network structures. Different from directly replacing the spatial up-sampling with the FourierUp in the baselines with spatial up-sampling, we need to encapsulate the FourierUp for the baselines without spatial up-sampling, in which a down-sampling operation is introduced to first reduce the resolution of features. We provide the detailed structures of the encapsulated FourierUp and the baselines with the FourierUp in the Figure 5 and supplementary material.

For the baselines with spatial up-sampling, we perform the comparison over four configurations:

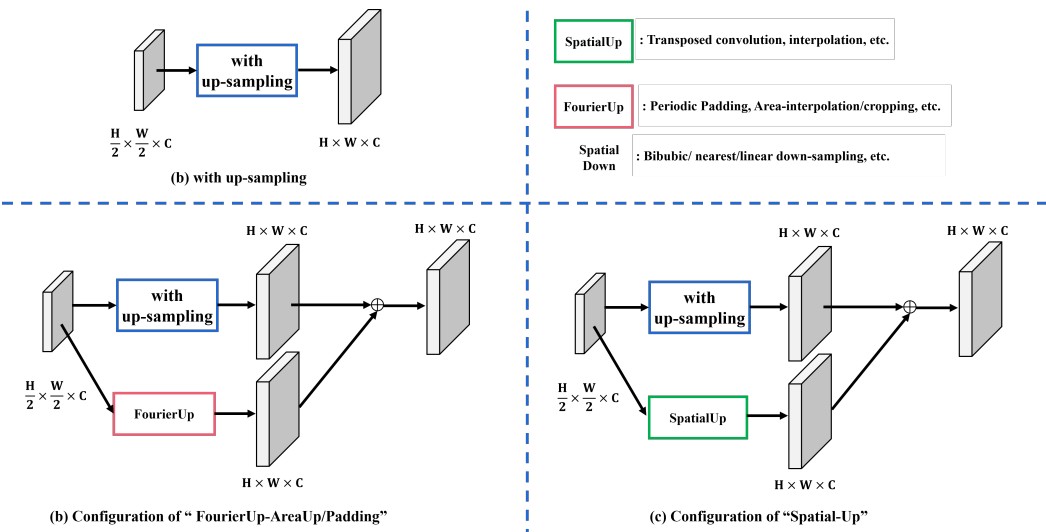

Figure 5: **The implementation details of FourierUp into the existing baselines with up-sampling.**

1) **Original**: the baseline without any changes;
2) **FourierUp-AreaUp**: replacing the original model's spatial up-sampling with the union of the Area-Interpolation variant of our FourierUp and the spatial up-sampling itself;
3) **FourierUp-Padding**: replacing the original model's spatial up-sampling operator with the union of the Periodic-Padding variant of our FourierUp and the spatial up-sampling itself;
4) **Spatial-Up**: replacing the variants of FourierUp in the settings of $2)/3)$ with the spatial up-sampling. For a fair comparison, we use the same number of trainable parameters as $2)/3)$.

For the baselines without spatial up-sampling, we perform the comparison over four configurations:

1) **Original**: the baseline without any changes;
2) **FourierUp-AreaUp**: replacing the original model's convolution with the encapsulated FourierUp that is equipped with the Area-Interpolation variant;
3) **FourierUp-Padding**: replacing the original model's convolution with the encapsulated FourierUp that is equipped with the Periodic-Padding variant;
4) **Spatial-Up**: replacing the the encapsulated FourierUp of the settings of $2)/3)$ with the spatial up-sampling. For a fair comparison, we use the same number of trainable parameters as $2)/3)$.

### 4.3 Comparison and Analysis

**Quantitative Comparison.** We perform the model performance comparison over different configurations, as described in implementation details. The quantitative results are presented in Tables 1 to 5 where the best and second best results are highlighted in bold and Underline. From the results, by integrating with our proposed two FourierUp variants, we can observe performance gain against the baselines across all the datasets in all tested tasks, suggesting the effectiveness of our approach. For example, for the PReNet of Table 1, "FourierUp-padding" and "FourierUp-AreaUp" obtain 0.83dB/0.65dB and 2.1dB/1.9dB PSNR gains than the "Original", 0.52dB/0.34dB and 1.7dB/1.5dB PSNR gains than "Spatial-Up" on the Rain200H and Rain200L datasets, respectively. Such results validate the effectiveness of our proposed FourierUp. The corresponding visualization consistently supports the analysis in Figure 6, where the FourierUp is capable of better maintaining the details.

**Qualitative Comparison.** Due to the limited space, we only report the visual results of the deraining/dehazing task in Figures 6 and 7 that can more clearly show the effectiveness of FourierUp. More results can be found in the supplementary materials. As shown, integrating the FourierUp with the original baseline achieves more visually pleasing results. Specifically, zooming-in the red box

region of Figures 6 and 7, the model equipped with the FourierUp is capable of better recovering the texture details while removing the rain/hazy effect.

Table 1: **Quantitative comparisons of image de-raining.**

| Model | Configurations | Rain200H | | Rain200L | |
|---|---|---|---|---|---|
| | | PSNR | SSIM | PSNR | SSIM |
| LPNet | Original | 22.907 | 0.775 | 32.461 | 0.947 |
| | Spatial-Up | 22.956 | 0.777 | 32.522 | 0.950 |
| | FourierUp-AreaUp | 22.163 | 0.783 | 32.681 | 0.954 |
| | FourierUp-Padding | **23.295** | **0.786** | **32.835** | **0.956** |
| PReNet | Original | 29.041 | 0.891 | 37.802 | 0.981 |
| | Spatial-Up | 29.357 | 0.901 | 38.271 | 0.985 |
| | FourierUp-AreaUp | 29.690 | 0.903 | 39.776 | 0.985 |
| | FourierUp-Padding | **29.871** | **0.908** | **39.971** | **0.987** |

Table 2: **Comparison over image dehazing.**

| Model | configurations | PSNR | SSIM |
|---|---|---|---|
| AODNet | Original | 18.80 | 0.834 |
| | Spatial-Up | 18.91 | 0.838 |
| | FourierUp-AreaUp | 19.16 | 0.843 |
| | FourierUp-Padding | **19.35** | **0.847** |
| MSBDN | Original | 33.79 | 0.984 |
| | Spatial-Up | 33.90 | 0.984 |
| | FourierUp-AreaUp | 34.21 | 0.985 |
| | FourierUp-Padding | **34.35** | **0.987** |

Table 3: **Comparison over object detection.**

| Model | Methods | AP50 | mAP |
|---|---|---|---|
| Faster RCNN | Original | 79.13 | 79.10 |
| | Spatial-Up | 79.14 | 79.10 |
| | FourierUp-AreaUp | 79.16 | 79.13 |
| | FourierUp-Padding | **79.19** | **79.15** |
| YOLO-V3 | Original | 81.68 | 81.63 |
| | Spatial-Up | 81.68 | 81.63 |
| | FourierUp-AreaUp | 81.70 | 81.65 |
| | FourierUp-Padding | **81.72** | **81.68** |

Table 4: **Quantitative comparisons of medical image segmentation.**

| Model | Configurations | synapse | | CANDI | |
|---|---|---|---|---|---|
| | | DSC ↑ | HD95 ↓ | DSC ↑ | HD95 ↓ |
| U-Net | Original | 76.85 | 39.70 | 86.50 | 3.946 |
| | Spatial-Up | 76.94 | 38.59 | 86.59 | 3.826 |
| | FourierUp-AreaUp | 77.25 | 36.02 | 86.63 | 3.751 |
| | FourierUp-Padding | **77.37** | **35.86** | **86.70** | **3.327** |
| Att-UNet | Original | 77.77 | 36.02 | 86.29 | 5.601 |
| | Spatial-Up | 77.85 | 35.91 | 86.35 | 5.588 |
| | FourierUp-AreaUp | 78.11 | 34.54 | 86.50 | 4.851 |
| | FourierUp-Padding | **78.34** | **34.29** | **86.64** | **4.833** |

Table 5: **Quantitative comparisons of pan-sharpening.**

| Model | Configurations | WorldView-II | | | | GaoFen2 | | | |
|---|---|---|---|---|---|---|---|---|---|
| | | PSNR↑ | SSIM↑ | SAM↓ | ERGAS↓ | PSNR↑ | SSIM↑ | SAM↓ | EGAS↓ |
| PANNET | Original | 40.817 | 0.963 | 0.025 | 1.055 | 43.066 | 0.968 | 0.018 | 0.855 |
| | Spatial-Up | 40.988 | 0.963 | 0.025 | 1.031 | 43.897 | 0.973 | 0.018 | 0.737 |
| | FourierUp-AreaUp | 41.167 | 0.963 | 0.024 | 1.010 | 45.964 | 0.979 | 0.015 | 0.653 |
| | FourierUp-Padding | **41.288** | **0.965** | **0.024** | **1.007** | **46.145** | **0.982** | **0.012** | **0.622** |
| DCFNET | Original | 40.276 | 0.968 | 0.028 | 1.051 | 42.986 | 0.967 | 0.019 | 0.858 |
| | Spatial-Up | 40.319 | 0.968 | 0.028 | **1.046** | 43.157 | 0.970 | 0.017 | 0.850 |
| | FourierUp-AreaUp | 40.484 | 0.968 | 0.025 | 1.115 | 43.881 | 0.979 | 0.014 | 0.829 |
| | FourierUp-Padding | **40.546** | **0.968** | **0.025** | 1.102 | **44.153** | **0.981** | **0.014** | **0.765** |

## 5  Limitations

First, the more comprehensive experiments on broader computer vision tasks (*e.g.*, image de-noising and image de-blurring) have not been explored. Second, the deep Fourier Up-sampling integrated with spatial up-sampling will increase the model parameter numbers. This is negligible at the significant performance gain at fewer parameter increase. Note that, the focus of this work is beyond designing a plug-and-play module to integrate it into existing networks for further performance gain. This work

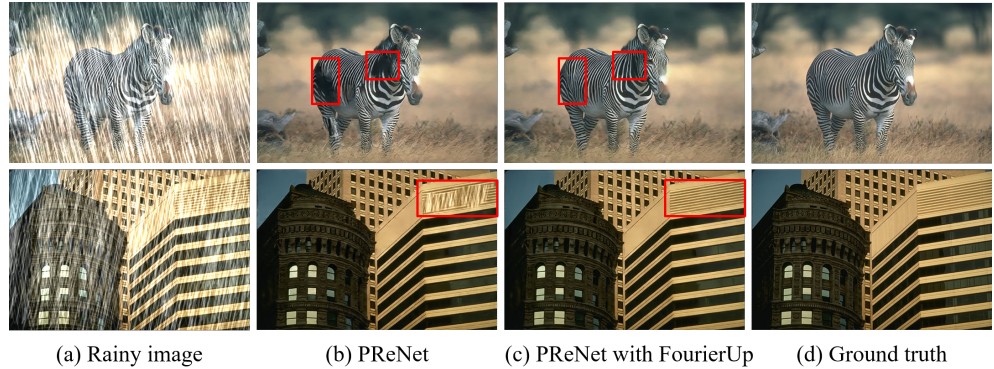

| (a) Rainy image | (b) PReNet | (c) PReNet with FourierUp | (d) Ground truth |

Figure 6: **Visual comparison of PReNet on the Rain200H.**

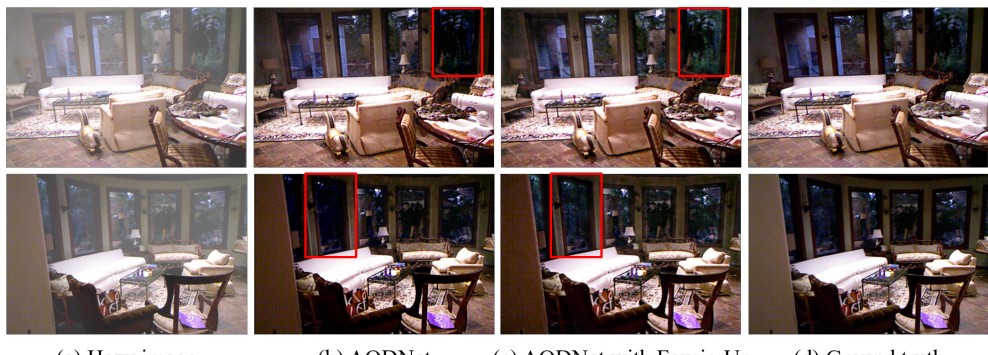

| (a) Hazy image | (b) AODNet | (c) AODNet with FourierUp | (d) Ground truth |

Figure 7: **Visual comparison of AODNet on the SOTS.**

also provides a powerful scale change choice of the up-sampling operator pool when developing a new model from start.

## 6 Conclusion

In this paper, we have proposed a deep Fourier up-sampling to explore the possibility of the up-sampling in the Fourier domain, which provides the key insight for the multi-scale Fourier pattern modeling. We theoretically demonstrate that our designs of Fourier up-sampling are feasible. It is appealing that the proposed FourierUp is a generic operator, thus being directly integrated with existing networks. Extensive experiments demonstrate the effectiveness of our method. We believe the FourierUp has the potential to advance broader computer vision tasks, *e.g.*, image/video super-resolution and image/video in-painting.

## Broader Impact

Our work shows the promising capability of up-sampling in the Fourier domain for computer vision algorithms through two novel designs with theoretical proofs. Using our deep Fourier up-sampling with negligible computational cost will improve the performance of neural networks and facilitate the development of AI in real-world applications. However, the efficacy of our method may raise potential concerns when it is improperly used. For example, the safety of the applications of our method in real-world applications may not be guaranteed. We will investigate the robustness and effectiveness of our method in broader real-world applications.

## Acknowledgements

We gratefully acknowledge the support of the Major Key Project of PCL (PCL2021A12), MindSpore, CANN, the JKW Research Funds under Grant 20-163-14-LZ-001-004-01 and Ascend AI Processor used for this research. Chongyi Li and Chen Change Loy are supported by the RIE2020 Industry Alignment Fund – Industry Collaboration Projects (IAF-ICP) Funding Initiative, as well as cash and in-kind contribution from the industry partner(s).

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
