# Deep Fourier Up-Sampling
# (Supplementary Material)

**Man Zhou**[1,2]*, **Hu Yu**[2]*, **Jie Huang**[2], **Feng Zhao**[2],
**Jinwei Gu**[6], **Chen Change Loy**[3], **Deyu Meng**[4,5], **Chongyi Li**[3†]

[1]Hefei Institute of Physical Science, Chinese Academy of Sciences, China
[2]University of Science and Technology of China, China
[3]S-Lab, Nanyang Technological University, Singapore
[4]Xi'an Jiaotong University, China
[5]Pazhou Laboratory (Huangpu), China
[6]SenseBrain Technology (SenseTime Research USA), USA
{manman,yuhu520,hj0117}@mail.ustc.edu.cn, fzhao956@ustc.edu.cn,
gujinwei@sensebrain.site, dymeng@mail.xjtu.edu.cn
{ccloy,chongyi.li}@ntu.edu.sg
https://li-chongyi.github.io/FourierUp_files/

This supplementary document is organized as follows:

Section 1 provides the alternative solution of Deep Fourier Up-Sampling variant: "Corner Interpolation". Due to the page limits, we only present two variants in main manuscript . Specifically, the theoretical evidences and module construction for FourierUp of "Corner Interpolation Variant" are reported in Section 1.1 and Section 1.2 respectively. The experimental results of "Corner Interpolation Variant" are presented in Section 1.3.

Section 2 provides the implementation details as shown in Fig. 4 and Fig. 5. To be specific, the configurations including "Original", "FourierUp-AreaUp", "FourierUp-Padding", and "Spatial-Up" in "Implementation Details" of main manuscript are illustrated.

Section 3 provides more quantitative and qualitative experimental results.

Section 4 provides more visualization of feature maps between the baselines and the ones integrated with the proposed "FourierUp".

## 1   Deep Fourier Up-Sampling Variant: "Corner Interpolation"

We first illustrate the alternative solution "Corner Interpolation" of Deep Fourier Up-Sampling, and then present the theoretical evidences, finally detail the corresponding module construction in the peso-code. The illustration of "Corner Interpolation" is shown in Fig. 1.

**Theorem-1.** Suppose the shifted $F_G^{shift}$ of the Fourier map $G \in \mathbb{R}^{M \times N}$ as

$$F_G^{shift}(u', v') = G(u - \frac{M}{2}, v - \frac{N}{2}),\tag{1}$$

where $u' = 0, 1, \ldots, M - 1$ and $v' = 0, 1, \ldots, N - 1$, it holds that the inverse Fourier transform $f_g^{shift}$ of $F_G^{shift}$

$$f_g^{shift}(x, y) = (-1)^{(x+y)} g(x, y),\tag{2}$$

where $x = 0, 1, \ldots, M - 1$ and $y = 0, 1, \ldots, N - 1$.

---

*Man Zhou and Hu Yu contribute equally.

†Corresponding author: Chongyi Li.

36th Conference on Neural Information Processing Systems (NeurIPS 2022).

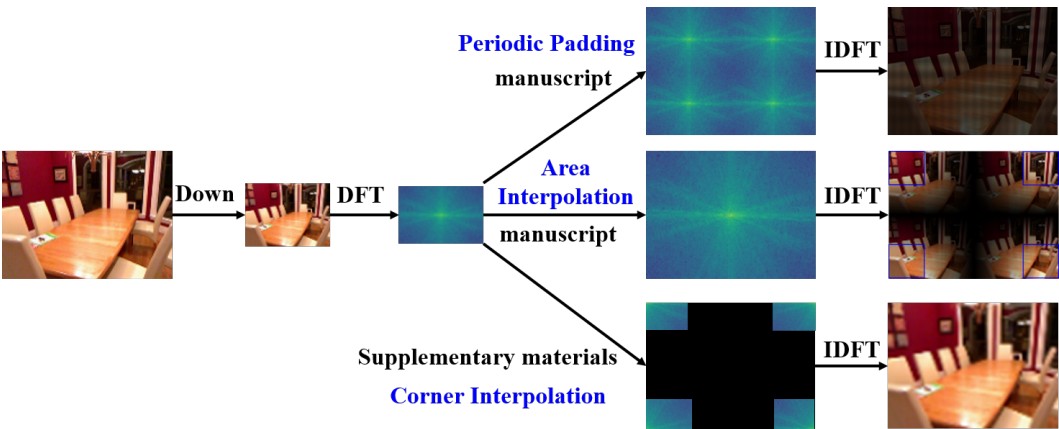

Figure 1: **An illustration of the proposed deep Fourier Up-Sampling.** It has three alternative variants: Periodic Padding and Area Interpolation/Cropping, as illustrated in main manuscript . The alternative solution "Corner Interpolation" is shown in supplementary materials.

## 1.1 Theoretical Evidences for FourierUp of "Corner Interpolation Variant"

For a spatial map $G(u, v) \in \mathbb{R}^{M \times N}$, we denote its 2-times up-sampled corner interpolation as $F_G^{cor}(u, v) \in \mathbb{R}^{2M \times 2N}$. Denote $G(u, v) \in \mathbb{R}^{M \times N}$, $F_g^{cor}(u, v) \in \mathbb{R}^{2M \times 2N}$, $F_g^{shiftcor}(u, v) = F_G^{cor}(u - M, v - N) \in \mathbb{R}^{2M \times 2N}$ as the corresponding Fourier transforms of $g(x, y)$, $f_g^{cor}(x, y)$, and $f_g^{shiftcor}(x, y)$ respectively.

The 2D Inverse Discrete Fourier transform (IDFT) of $G(u, v)$ can be written as

$$g(x, y) = \frac{1}{MN} \sum_{u=0}^{M-1} \sum_{v=0}^{N-1} G(u, v) e^{j2\pi(\frac{ux}{M} + \frac{vy}{N})}. \tag{3}$$

We up-sample $G(u, v) \in \mathbb{R}^{M \times N}$ to get $F_G^{cor}(u, v) \in \mathbb{R}^{2M \times 2N}$ by corner interpolation. Specifically, the corner interpolation is shown in Fig. 3. For convenience, we infer the inverse Fourier transform of $F_g^{shiftcor}(u, v)$ as

$$
\begin{aligned}
f_G^{shiftcor}(x, y) &= \frac{1}{4MN} \sum_{u=0}^{2M-1} \sum_{v=0}^{2N-1} F_G^{cor}(u, v) e^{j2\pi(\frac{ux}{2M} + \frac{vy}{2N})} \\
&= \frac{1}{4MN} \sum_{u=0}^{M-1} \sum_{v=0}^{N-1} F_G^{AI}(2u, 2v) e^{j2\pi(\frac{2ux}{2M} + \frac{2vy}{2N})} + \frac{1}{4MN} \sum_{u=0}^{M-1} \sum_{v=0}^{N-1} F_G^{AI}(2u+1, 2v) e^{j2\pi(\frac{(2u+1)x}{2M} + \frac{2vy}{2N})} \\
&= \frac{1}{4MN} \sum_{u=\frac{M}{2}}^{\frac{3M}{2}} \sum_{v=\frac{N}{2}}^{\frac{3N}{2}} F_G^{cor}(u, v) e^{j2\pi(\frac{2ux}{2M} + \frac{(2v+1)y}{2N})} \\
&= \frac{1}{4MN} \sum_{u=\frac{M}{2}}^{\frac{3M}{2}} \sum_{v=\frac{N}{2}}^{\frac{3N}{2}} G(u - \frac{M}{2}, v - \frac{N}{2}) e^{j2\pi(\frac{2ux}{2M} + \frac{(2v+1)y}{2N})}.
\end{aligned}
\tag{4}
$$

Let $u' = u - \frac{M}{2}$ and $v' = v - \frac{N}{2}$, the equation (4) is transformed as

$$
\begin{aligned}
f_G^{shiftcor}(x,y) &= \frac{1}{4MN} \sum_{u=\frac{M}{2}}^{\frac{3M}{2}} \sum_{v=\frac{N}{2}}^{\frac{3N}{2}} G(u - \frac{M}{2}, v - \frac{N}{2}) e^{j2\pi(\frac{2ux}{2M} + \frac{(2v+1)y}{2N})} \\
&= \frac{1}{4MN} \sum_{u'=0}^{M-1} \sum_{v'=0}^{N-1} G(u',v') e^{j2\pi(\frac{(u'+M/2)x}{2M} + \frac{(v'+N/2)y}{2N})} \\
&= \frac{1}{4MN} \sum_{u'=0}^{M-1} \sum_{v'=0}^{N-1} G(u',v') e^{j2\pi(\frac{u(x/2)}{M} + \frac{v(y/2)}{N})} e^{j\pi(\frac{x}{2} + \frac{y}{2})} \\
&= \frac{1}{4} g(\frac{x'}{2}, \frac{y'}{2}) e^{j\pi(\frac{x'}{2} + \frac{y'}{2})},
\end{aligned}
\tag{5}
$$

where $x' = 2x$ and $y' = 2y$, $x = 0, 1, \ldots, M-1$ and $y = 0, 1, \ldots, N-1$. Similarly, we can write $A(x,y) = e^{j\pi(\frac{x'}{2} + \frac{y'}{2})}$ as $|A(x,y)| = 1$ for any integer $x$, $y$. Recall **Theorem-1**, we can infer $f_G^{cor}(x,y)$ as

$$
\begin{aligned}
f_G^{cor}(x,y) &= (-1)^{(x+y)} f_G^{shiftcor}(x,y) \\
&= g(\frac{x'}{2}, \frac{y'}{2}) e^{j\pi(\frac{x'}{2} + \frac{y'}{2})} \frac{(-1)^{(x+y)}}{4}.
\end{aligned}
\tag{6}
$$

We can find that the even points in $f_G^{cor}(2x, 2y)$ are equal to the corresponding point in $\frac{g(x,y)}{4}$. The odd points have no definitions and are obtained by interpolation, acting as the low-pass filtering in spatial domain.

**Theorem-2.** Suppose the corner interpolated $F_G^{cor}$ of the Fourier map $G \in \mathbb{R}^{M \times N}$, it holds that the inverse Fourier transform $f_g^{cor}$ of $F_G^{cor}$

$$
f_g^{cor}(x,y) = g(\frac{x'}{2}, \frac{y'}{2}) e^{j\pi(\frac{x'}{2} + \frac{y'}{2})} \frac{(-1)^{(x+y)}}{4},
\tag{7}
$$

where $x' = 2x$ and $y' = 2y$, $x = 0, 1, \ldots, M-1$ and $y = 0, 1, \ldots, N-1$.

## 1.2 FourierUp Module Design of "Corner Interpolation Variant"

Recalling **Theorem-1** and **Theorem-2**, we correspondingly propose the Fourier up-sampling module, called Corner Interpolation Variant.

**Corner Interpolation Up-Sampling.** The pseudo-code of the Corner Interpolation Up-sampling is shown in Fig. 2. Specifically, given an image $X \in \mathbb{R}^{H \times W \times C}$, we first adopt the Fourier transform $FFT(X)$ to obtain its amplitude component A and phase component P. We then perform the Corner Interpolation over A and P two times in both the $H$ and $W$ dimensions, as illustrated in Fig. 3, to form the padded A_pep and P_pep. Such up-sampling maps are then fed into two independent convolution modules with a kernel size of $1 \times 1$ and followed by the inverse Fourier transform $iFFT(.)$ to project the padded ones back to the spatial domain.

Note that albeit being designed on the basis of strict theories, both constructed spectral up-sampling modules contain certain approximations, like a learnable $1 \times 1$ convolution operator instead of strictly $1/4$ as described in Theorem-1 of main manuscript , and an approximation cropping to preserve the map corners instead of accurate $A$ mapping as proved in Theorem-2 of main manuscript and Theorem-2 of supplementary materials. Such strategy makes the proposed modules able to be more easily implemented and more flexibly represent real data spectral structures. It is worth noting that this should be the first attempt for constructing easy equitable spectral upsampling modules, and hope it would inspire more effective and rational ones from more spectral perspectives.

## 1.3 Comparison and Analysis

We perform the model performance comparison over different configurations, as described in implementation details of main manuscript . The quantitative results are presented in Tables 1 to 2 where the best and second best results are highlighted in bold and underline. From the results, by

```python
def FourierUp_CornerInterpolation(X):
# X: input with shape [N, C, H, W]
# A and P are the amplitude and phase
    A, P = FFT(X)

    # Fourier up-sampling transform rules
    A_aip = Corner-Interpolation(A)
    P_aip = Corner-Interpolation(P)
    A_aip = Convs_1x1(A_aip)
    P_aip = Convs_1x1(P_aip)

    # Inverse Fourier transform
    Y = iFFT(A_aip, P_aip)

    Return Y #[N, C, 2H, 2W]

def Corner-Interpolation(X):
# X: input with shape [N, C, H, W]
# A and P are the amplitude and phase
    r, c = X.shape(2), X.shape(3)

    I_Mup=torch.zeros((N, C, 2*H, 2*W))
    I_Pup=torch.zeros((N, C, 2*H, 2*W))

    if r%2==1:#odd
        ir1,ir2=r//2+1,r//2+1
    else: #even
        ir1,ir2=r//2+1,r//2
    if c%2==1:#odd
        ic1,ic2=c//2+1,c//2+1
    else: #even
        ic1,ic2=c//2+1,c//2

    A_aip[:,:,:ir1,:ic1]=A[:,:,:ir1,:ic1]
    A_aip[:,:,:ir1,ic2+c:]=A[:,:,:ir1,ic2:]
    A_aip[:,:,ir2+r:,:ic1]=A[:,:,ir2:,:ic1]
    A_aip[:,:,ir2+r:,ic2+c:]=A[:,:,ir2:,ic2:]

    if r%2==0:#even
        A_aip[:,:,ir2,:]=A_aip[:,:,ir2,:]*0.5
        A_aip[:,:,ir2+r,:]=A_aip[:,:,ir2+r,:]*0.5
    if c%2==0:#even
        A_aip[:,:,:,ic2]=A_aip[:,:,:,ic2]*0.5
        A_aip[:,:,:,ic2+c]=A_aip[:,:,:,ic2+c]*0.5

    P_aip[:,:,:ir1,:ic1]=P[:,:,:ir1,:ic1]
    P_aip[:,:,:ir1,ic2+c:]=P[:,:,:ir1,ic2:]
    P_aip[:,:,ir2+r:,:ic1]=P[:,:,ir2:,:ic1]
    P_aip[:,:,ir2+r:,ic2+c:]=P[:,:,ir2:,ic2:]

    if r%2==0:#even
        I_Pup[:,:,ir2,:]=P_aip[:,:,ir2,:]*0.5
        I_Pup[:,:,ir2+r,:]=P_aip[:,:,ir2+r,:]*0.5
    if c%2==0:#even
        P_aip[:,:,:,ic2]=P_aip[:,:,:,ic2]*0.5
        P_aip[:,:,:,ic2+c]=P_aip[:,:,:,ic2+c]*0.5

    Return A_aip, P_aip
```

Figure 2: **Pseudo-code of the variant of the proposed deep Fourier up-sampling: Corner interpolation variant.**

Table 1: **Quantitative comparisons of image de-raining.**

| Model | Configurations | Rain200H | | Rain200L | |
|---|---|---|---|---|---|
| | | PSNR | SSIM | PSNR | SSIM |
| LPNet | Original | 22.907 | 0.775 | 32.461 | 0.947 |
| | Spatial-Up | 22.956 | 0.777 | 32.522 | 0.950 |
| | FourierUp-AreaUp | 22.163 | 0.783 | 32.681 | 0.954 |
| | FourierUp-Corner | 22.291 | 0.786 | 32.678 | 0.954 |
| | FourierUp-Padding | **23.295** | **0.786** | **32.835** | **0.956** |
| PReNet | Original | 29.041 | 0.891 | 37.802 | 0.981 |
| | Spatial-Up | 29.357 | 0.901 | 38.271 | 0.985 |
| | FourierUp-AreaUp | 29.690 | 0.903 | 39.776 | 0.985 |
| | FourierUp-Corner | 29.866 | 0.908 | 39.970 | 0.987 |
| | FourierUp-Padding | **29.871** | **0.908** | **39.971** | **0.987** |

Table 2: **Quantitative comparisons of pan-sharpening.**

| Model | Configurations | WorldView-II | | | | GaoFen2 | | | |
|---|---|---|---|---|---|---|---|---|---|
| | | PSNR↑ | SSIM↑ | SAM↓ | ERGAS↓ | PSNR↑ | SSIM↑ | SAM↓ | EGAS↓ |
| PANNET | Original | 40.817 | 0.963 | 0.025 | 1.055 | 43.066 | 0.968 | 0.018 | 0.855 |
| | Spatial-Up | 40.988 | 0.963 | 0.025 | 1.031 | 43.897 | 0.973 | 0.018 | 0.737 |
| | FourierUp-AreaUp | 41.167 | 0.963 | 0.024 | 1.010 | 45.964 | 0.979 | 0.015 | 0.653 |
| | FourierUp-Corner | 41.286 | 0.965 | 0.024 | 1.007 | 46.137 | 0.981 | 0.012 | 0.631 |
| | FourierUp-Padding | **41.288** | **0.965** | **0.024** | **1.007** | **46.145** | **0.982** | **0.012** | **0.622** |
| DCFNET | Original | 40.276 | 0.968 | 0.028 | 1.051 | 42.986 | 0.967 | 0.019 | 0.858 |
| | Spatial-Up | 40.319 | 0.968 | 0.028 | **1.046** | 43.157 | 0.970 | 0.017 | 0.850 |
| | FourierUp-AreaUp | 40.484 | 0.968 | 0.025 | 1.115 | 43.881 | 0.979 | 0.014 | 0.829 |
| | FourierUp-Corner | 40.539 | 0.968 | 0.027 | 1.105 | 44.139 | 0.979 | 0.014 | 0.771 |
| | FourierUp-Padding | **40.546** | 0.968 | **0.025** | 1.102 | **44.153** | **0.981** | **0.014** | **0.765** |

integrating with the FourierUp variant "Corner-interpolation", we can observe performance gain against the baselines across all the datasets in two representative tasks: image de-raining and pan-sharpening, suggesting the effectiveness of our approach. For example, for the PReNet of Table 1, "FourierUp-Corner" obtains 0.82dB/2.1dB PSNR gains than the "Original", 0.51dB/1.5dB PSNR gains than "Spatial-Up" on the Rain200H and Rain200L datasets, respectively. Such results validate the effectiveness of our proposed FourierUp.

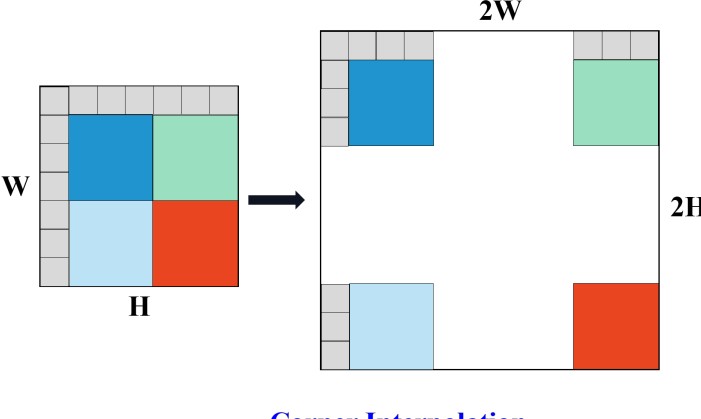

**Corner Interpolation**

Figure 3: **The illustrations of corner interpolation implementation in Fig. 1.** The gray parts represent a row/column pixels while the remaining color parts are evenly divided.

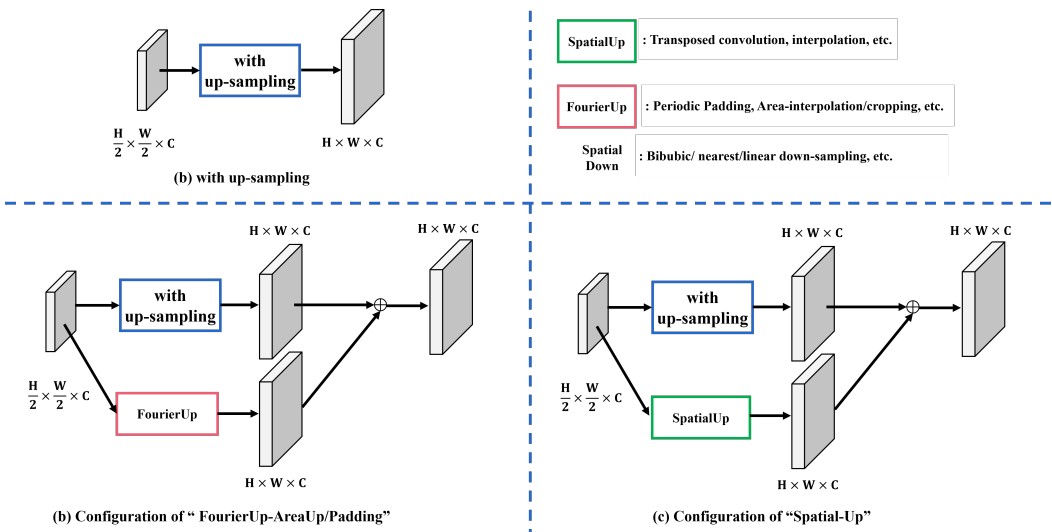

Figure 4: **The illustrations of corner interpolation over the baselines with up-sampling in Figure 1.** The gray parts represent a row/column pixels while the remaining color parts are evenly divided.

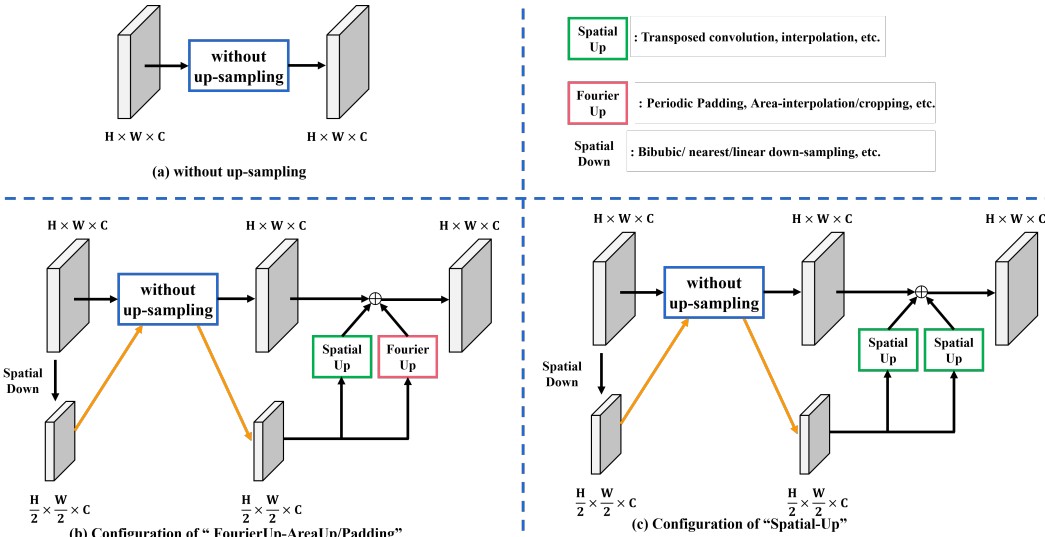

Figure 5: **The illustrations of corner interpolation over the baselines without up-sampling in Figure 1.** The gray parts represent a row/column pixels while the remaining color parts are evenly divided.

## 2   Implementation Details

Regarding the above competitive baselines, they can be divided into two categories: one with spatial up-sampling and another one without spatial up-sampling. We provide the detailed structures of the encapsulated FourierUp (detailed in Fig. 5(b)) and the baselines with the FourierUp in Fig. 5 and Fig. 4.

For the baselines with spatial up-sampling, we perform the comparison over four configurations:

1) **Original**: the baseline without any changes;
2) **FourierUp-AreaUp in Fig. 4(b)**: replacing the original model's spatial up-sampling with the union of the Area-Interpolation variant of our FourierUp and the spatial up-sampling itself;

3) **FourierUp-Padding in Fig. 4(b)**: replacing the original model's spatial up-sampling operator with the union of the Periodic-Padding variant of our FourierUp and the spatial up-sampling itself;
4) **FourierUp-Corner in Fig. 4(b)**: replacing the original model's spatial up-sampling operator with the union of the Corner-Interpolation variant of our FourierUp and the spatial up-sampling itself;
5) **Spatial-Up in Fig. 4(c)**: replacing the variants of FourierUp in the settings of $2)/3)$ with the spatial up-sampling. For a fair comparison, we use the same number of trainable parameters as $2)/3)$.

For the baselines without spatial up-sampling, we perform the comparison over four configurations:

1) **Original**: the baseline without any changes;
2) **FourierUp-AreaUp in Fig. 5(b)**: replacing the original model's convolution with the encapsulated FourierUp that is equipped with the Area-Interpolation variant;
3) **FourierUp-Padding in Fig. 5(b)**: replacing the original model's convolution with the encapsulated FourierUp that is equipped with the Periodic-Padding variant;
4) **FourierUp-Corner in Fig. 5(b)**: replacing the original model's convolution with the encapsulated FourierUp that is equipped with the Corner-Interpolation variant;
5) **Spatial-Up in Fig. 5(c)**: replacing the the encapsulated FourierUp of the settings of $2)/3)$ with the spatial up-sampling. For a fair comparison, we use the same number of trainable parameters as $2)/3)$.

## 3 Experiments

**Quantitative Comparison.** We adopt the pan-sharpening, the representative task of guided image super-resolution, for evaluations. Due to the page limits, the results over WorldView III have not been presented in main manuscript. We also employ two different network designs for validation, including PANNET without up-sampling operator and DCFNET with up-sampling operator.

We perform the model performance comparison over different configurations, as described in implementation details. The quantitative results are presented in Table 3 where the best and second best results are highlighted in bold and underline. From the results, by integrating with our proposed FourierUp variants, we can observe performance gain against the baselines across all the datasets, suggesting the effectiveness of our approach. For example, for the PANNET of Table 3, "FourierUp-padding", "FourierUp-AreaUp" and "FourierUp-Corner" obtain the performance gains than the "Original" and "Spatial-Up" on the WorldView-III datasets, respectively. Such results validate the effectiveness of our proposed FourierUp. The corresponding visualization consistently supports the analysis in Fig. 8, where the FourierUp is capable of better maintaining the details.

**Qualitative Comparison.** Due to the limited space, we only report the visual results of the de-raining/dehazing task in main manuscript. We report more visual results in the supplementary materials. As shown, integrating the FourierUp with the original baseline achieves more visually pleasing results. Specifically, zooming-in the red box arrows of Fig. 6 and 7, the model equipped with the FourierUp is capable of better recovering the texture details while removing the rain/haze effect.

## 4 Comparison in Feature Space

In this section, we present more visualization results of feature maps, demonstrating the effectiveness of the FourierUp. Fig. 9 and Fig. 10 present the representative example in the PreNet over image de-raining dataset-Rain200H. As can be seen, the PreNet integrated with our proposed FourierUp is capable of better distinguishing and disentangling the rain features and background features, thus improving the model performance while the original PreNet suffers from severe feature entanglement over rain streaks and background.

Table 3: **Quantitative comparisons of pan-sharpening.**

| Model | Configurations | WorldView-III | | | |
| --- | --- | --- | --- | --- | --- |
| | | PSNR↑ | SSIM↑ | SAM↓ | ERGAS↓ |
| PANNET | Original | 29.68 | 0.907 | 0.085 | 3.426 |
| | Spatial-Up | 29.71 | 0.907 | 0.085 | 3.426 |
| | FourierUp-AreaUp | 30.05 | 0.915 | 0.078 | 3.253 |
| | FourierUp-Padding | **30.24** | **0.918** | **0.077** | **3.187** |
| DCFNET | Original | 29.47 | 0.907 | 0.089 | 3.536 |
| | Spatial-Up | 29.51 | 0.907 | 0.088 | 3.536 |
| | FourierUp-AreaUp | 29.69 | 0.913 | 0.085 | 3.326 |
| | FourierUp-Padding | **29.85** | **0.914** | **0.078** | 3.219 |

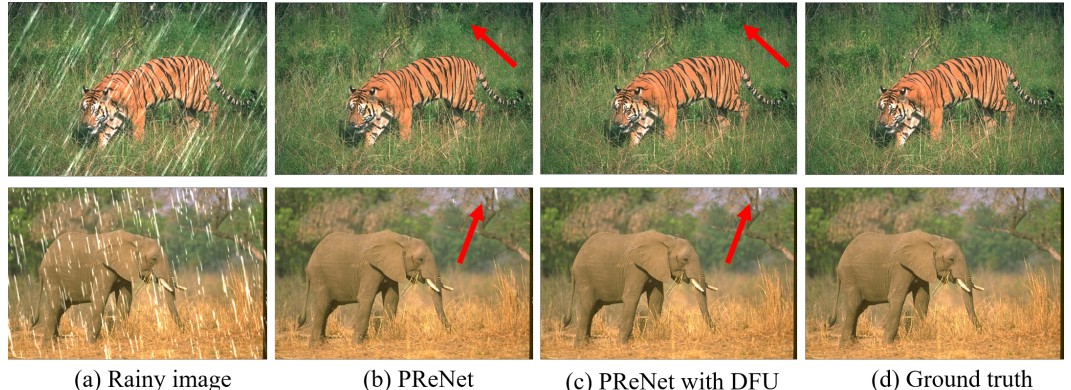

| (a) Rainy image | (b) PReNet | (c) PReNet with DFU | (d) Ground truth |
| --- | --- | --- | --- |

Figure 6: **Visual comparison of PReNet on the Rain200L.**

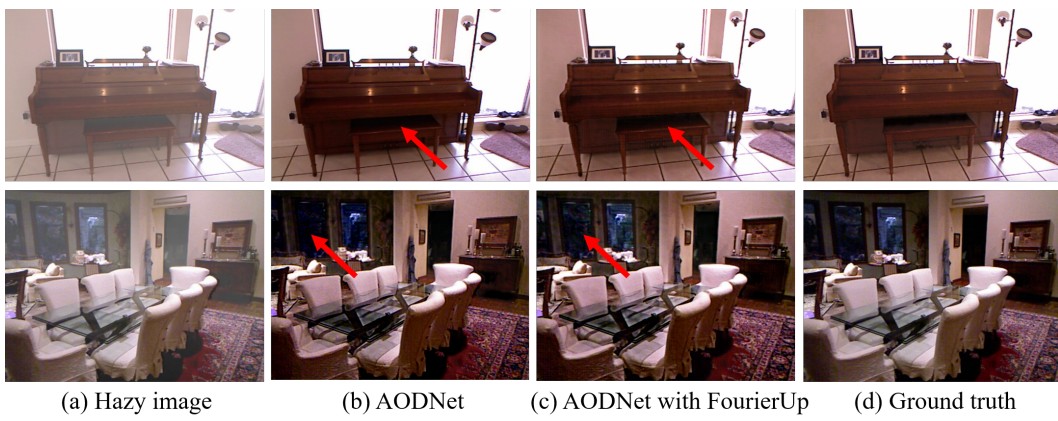

| (a) Hazy image | (b) AODNet | (c) AODNet with FourierUp | (d) Ground truth |
| --- | --- | --- | --- |

Figure 7: **Visual comparison of AODNet on the SOTS.**

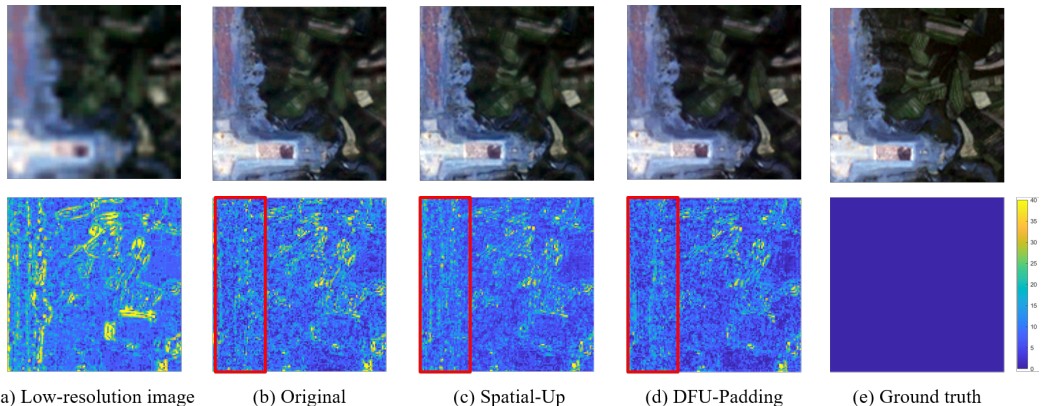

(a) Low-resolution image    (b) Original    (c) Spatial-Up    (d) DFU-Padding    (e) Ground truth

Figure 8: The visual comparison of PANNET over WorldView-II.

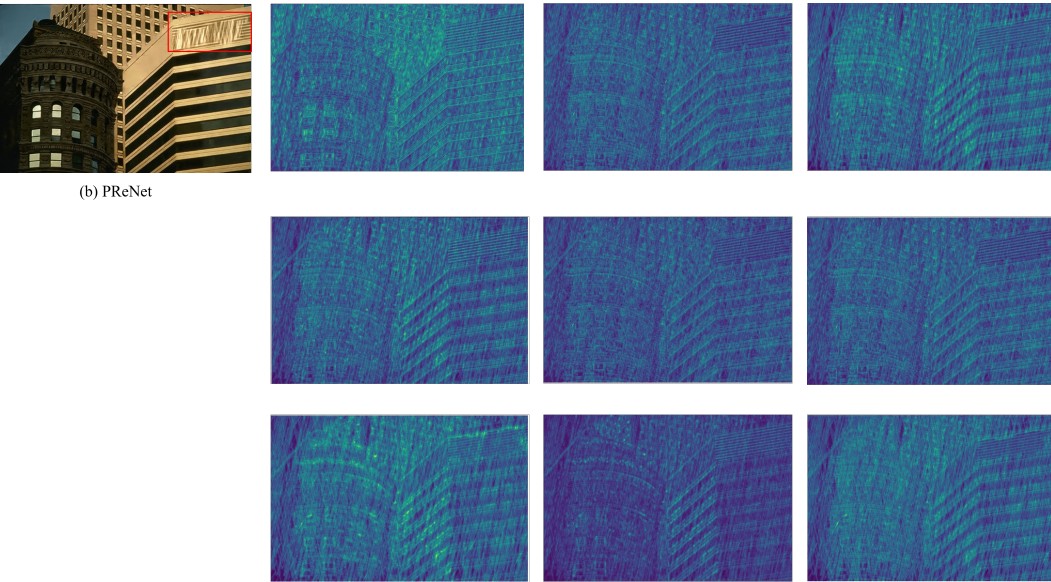

(b) PReNet

Figure 9: The visual feature maps comparison.

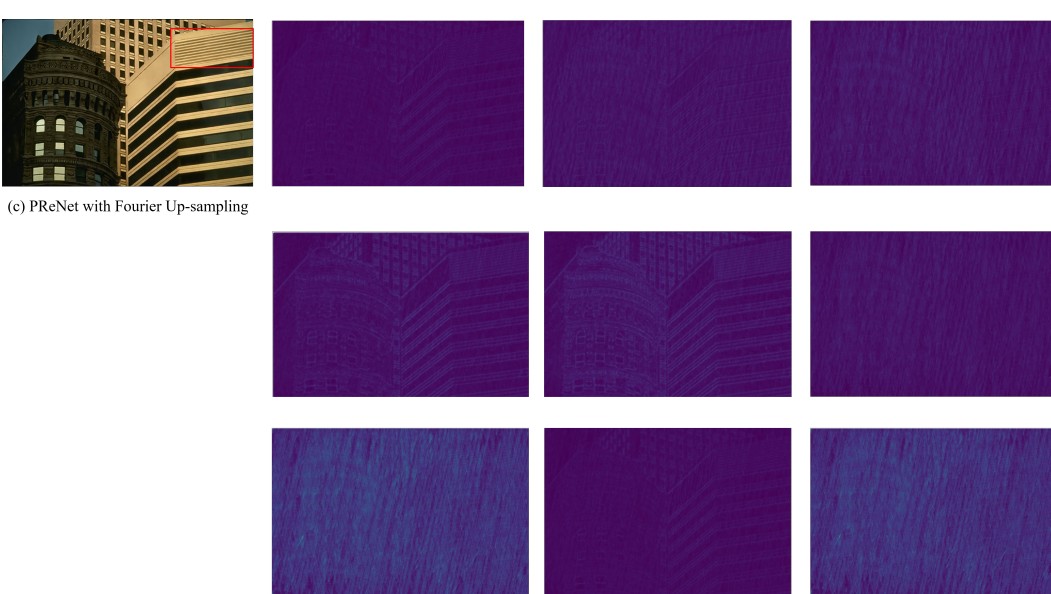

(c) PReNet with Fourier Up-sampling

Figure 10: The visual feature maps comparison.