# OpenReview forum: "Deep Fourier Up-Sampling"
_NeurIPS.cc/2022/Conference — NeurIPS 2022 Accept_

### Official Review · Reviewer_tPpX · 2022-06-24

**Rating:** 3
**Confidence:** 4
**Soundness:** 2 fair
**Presentation:** 1 poor
**Contribution:** 2 fair

**Summary:**

The authors propose an upsampling layer based on the Fourier transform and properties of the frequency domain. The motivation behind the work is quite vague, only resorting to the fact that the Fourier transform is a global operator rather than local.

**Questions:**

I would like the authors to clarify where the novelty lies in the proposed method and a more precise explanation of why we should expect gains with respect to existing techniques.

**Limitations:**

The authors addressed the limitation that two very important benchmarks (classification and denoising) are missing. Still, when proposing a very general technique, the body of proof required to demonstrate it is advantageous needs to be substantially larger than what is presented.

**Strengths And Weaknesses:**

The paper is generally poorly written, lacks a clear motivation, and the novelty is unclear.

Strengths:
- Frequency-based techniques for layer operations are an area of research that deserves more interest

Weaknesses:
- Poor presentation. Readability is very low, especially due to long derivations and proofs in the text rather than in the appendix. It is hard to grasp what the method does at a glance
- Novelty unclear. It seems that the work just exploits well known properties of the Fourier transform of upsampled signals. It is also unclear why the scheme should outperform any other technique
- Experiments show very marginal improvements and lack two very important problems: image classification and denoising

---

> ### Author Response · Authors · 2022-08-02
> **Response to Reviewer tPpX (part 2/2)**
>
> **2, two very important problems.**
>
> (1) To our best knowledge, existing representative classification networks, e.g., VGG, ResNet, and InceptionNet, don not adopt the up-sampling operators. Therefore, we cannot report the experiments over the image classification task.
>
> (2) As a common sense, image de-raining and  image de-noise follow the similar degradation formula:
>
> Y= X+N
>
> where Y and X indicate the degraded image and the latent clear version.  For image de-raining, N represents the rain effect. In terms of de-noise, N is the noise. Therefore, we only report the image de-raining task.
>
> As suggested, we have added the experiments for image denoising. Due to the limited time, we only perform the image de-noising experiments with regards to the representative baseline VDN [1] using the commonly-used SIDD benchmark [2]. The results are presented in this table. As presented, the original baseline obtains the performance gain in terms of PSNR and SSIM when equipped with all the proposed “Fourier Up-sampling” variants. The results are consistent with other vision tasks presented in our main paper.
>
> |        |       | Original | Spatial-Up | FourierUp-AreaUp | FourierUp-Padding |
> |:------:|:-----:|:--------:|:----------:|:----------------:|:-----------------:|
> |  VDN | PSNR |  39.23  |    39.31  |       39.55     |       39.67       |
> |        | SSIM |  0.971 |   0.971  |      0.973     |     0.973     |
>
> [1] Zongsheng Yue, Hongwei Yong, Qian Zhao, Lei Zhang, Deyu Meng. Variational Denoising Network: Toward Blind Noise Modeling and Removal. NeurIPS, 2019.
>
> [2] Abdelrahman Abdelhamed, Stephen Lin, and Michael S. Brown. A high-quality denoising dataset for smartphone cameras. In The IEEE Conference on Computer Vision and Pattern Recognition (CVPR), June 2018.
>
> **3, Marginal improvements.**
>
> In the manuscript, it can be found that our proposed Fourier Up-sampling achieves significant performance gain of different degrees over all the reported vision tasks. Compared with existing works, such improvements are significant, not marginal as these tasks are challenging. Therefore, according to previous studies, we would claim that these improvements are very important and meaningful when the computational burden introduced by our method is negligible.

---

> > ### Comment · Reviewer_tPpX · 2022-08-06
> > **Thank you for the response**
> >
> > I would like to thank the authors for their detailed response. However, I remain unconvinced about the overall significance of the work.
> >
> > In their rebuttal, the authors clarify the main idea behind moving upsampling ot the Fourier domain is enabling a global receptive field for the operation under the assumption that this would be better than more or less localized operations in the spatial domain. This motivation seems questionable for a couple of reasons. First, it is not obvious that the frequency space with its total loss of spatial coordinates provides a more effective domain for interpolation. Indeed, it has been known since scale-space models and space-frequency transforms like wavelets that it is hard to exploit self-similarity in the Fourier domain. Second, the authors rightly point out the extreme locality (and linearity) of transpose convolution,but this operation is to be considered outdated and causing artifacting. The more popular current approach of pixel shuffling does not suffer from these limitations to the same extent due to the fact that pixels can be derived from features exploiting the entire receptive field of the network (which can be quite large). Moreover, thank to its use of an arbitrary feature space it effectively implements a non-linear polyphase decomposition of the signal.
> >
> > Related to the issues of novelty and presentation, the derivation that takes most of page 4 is a standard result that can be found in any signal processing textbook. The result on page 5 is also a trivial property.
> >
> > Regarding, the experimental results, it seems that the authors use rather dated baselines (most are from 2018-2020). Even for the denoising experiment they graciously reported in the rebuttal, the VDN model from 2019 is currently outdated and outperformed by a number of methods. Due to this concern, there is no evidence that the proposed technique can work on actual state-of-the-art models and any gains observed in the paper might be due to the suboptimal designs of these older models.

---

> > > ### Author Response · Authors · 2022-08-08
> > > **Response to Reviewer tPpX**
> > >
> > > Thanks for your comment!
> > >
> > > I totally disagree with you. First, according to the spectral convolution theorem in Fourier theory, updating a single value in the spectral domain globally affects all original spatial data, which sheds light on design efficient neural architectures with non-local receptive field. Therefore, the proposed Fourier Up-sampling empowers the network the global information modeling ability in an effective manner. At the last count, in terms of Transposed convolution, it is the widely used up-sampling operator and not to be considered outdated. In addition, the pixel shuffle is not involved trainable parameters and coupled with pure convolution to perform the up-sampling action. However, the pure convolution and Transposed convolution is locally linear operator while our proposed Fourier Up-sampling is nonlinear, thus improving the model learning ability. Furthermore, in the line 39-42, we have claimed “However, the aforementioned methodologies only interact at a single resolution scale, and the spatial-Fourier interaction potential of multiple scales in the Fourier domain has not been investigated. The key to solving this problem lies in how to implement deep Fourier up-sampling for multi-scale Fourier pattern modeling.” To emphasize, our focus is to design the Fourier up-sampling for the multi-scale spatial-Fourier interaction.
> > >
> > > Second, the selected baselines are the representative works in the related tasks. In addition, as a general operators, implementing the proposed Fourier Up-sampling in the standard benchmarks is more fair. Further, as you suggested, we implement the proposed Fourier Up-sampling into the state-of-the-art image de-raining method SGCN [1] to conduct experimental validation on Rain200L and Rain200H datasets. As can be seen in this table, with the proposed Fourier Up-sampling operators, the performance of original SGCN is improved in terms of PSNR and SSIM and the corresponding variant achieves the best de-raining results.
> > >
> > > Finally, we will insert our proposed Fourier Up-sampling into the state-of-the-art benchmarks of other reported tasks in revision.
> > >
> > > |        |                   | Rain200L |       | Rain200H |       |
> > > |:------:|:-----------------:|:--------:|:-----:|:--------:|:-----:|
> > > | method |   configuration   |   PSNR   |  SSIM |   PSNR   |  SSIM |
> > > |        |      Original     |   37.65  | 0.976 |   29.13  | 0.895 |
> > > |  SGCN  |     Spatial-Up    |   37.86  | 0.982 |   29.47  | 0.903 |
> > > |        |  FourierUp-AreaUp |   39.57  | 0.985 |   29.82  | 0.908 |
> > > |        | FourierUp-Padding |   39.83  | 0.987 |   29.95  | 0.913 |
> > >
> > > [1] Xueyang Fu, et.al. Successive Graph Convolutional Network for Image De-raining. IJCV, 2022.

---

> ### Author Response · Authors · 2022-08-02
> **Response to Reviewer tPpX (part 1/2)**
>
> **1, Presentation and novelty.**
>
> In terms of presentation, we are encouraged by other reviewers who recognize the good presentation of this paper. Specifically, Reviewer-1 admits “The paper writing and explanations are good. Reviewer-2 and Reviewer-3 comment “easy to follow and clear writing”. In addition, w also provide detailed proofs and illustrations that help one understand our method and the specific implementation in the main paper and supplementary material.
>
> In terms of novelty, we wish to emphasize the contributions of our work again.
>
> 1) We propose a novel Deep Fourier Up-Sampling, which enables the integration of the features of different resolutions in the Fourier domain. To the best of our knowledge, this is the first thorough effort to explore the Fourier up-sampling for multi-scale modeling.
> 2)  The proposed FourierUp is a generic operator that can be directly integrated with the existing networks, which provides flexible plug-and-play.
> 3) Equipped with the theoretically feasible FourierUp, the networks achieve consistent performance improvement across multiple computer vision tasks, which suggests the potential of our method for refreshing the neural network designs in the Fourier domain.
>
> In addition, other reviewers also confirm our novelty and contributions. To be specific, Reviewer-1 highly praises our work “I actually had this kind of idea but never tried this. It is refreshing to see that this paper has done this. This work will drive the community in a certain direction”.  Reviewer-2 confirms our novelty: “Idea is interesting”. Reviewer-3 praises that “Interesting motivation: This work takes an interesting and challenging issue, and Theoretical proof: For the proposed Deep Fourier Up-Sampling, this work provides detailed and rigorous proof, making a theoretical sound solution. These proofs provide a new view to understand deep Fourier up-sampling and convince the readers.”
>
> Finally, in terms of the question “a more precise explanation of why we should expect gains with respect to existing techniques”, the underlying reasons why the FourierUp works better are summarized as follows:
>
> (a) First, our proposed “Fourier Up-Sampling” is performed in the frequency feature space where each factor connects with all the pixels in the spatial domain via Flourier transformation. In other words, learning in the frequency domain allows obtaining the image-wide receptive field that models the global contexts that are indispensable for vision tasks. Different from the existing spatial up-sampling strategies that only depend on local pixel attention for up-sampled reconstruction, our proposed “Fourier Up-Sampling” is performed in the frequency domain and thus fully explores the global dependency for up-sampling.
>
> (b) Second, the spatial up-sampling is implemented by Transposed convolution and various interpolation techniques and both of them are linear operators. Therefore, the spatial up-sampling also can be regarded as a linear transformation. Differently, our “Fourier Up-Sampling” is implemented in the Fourier space and the transposed components are processed by different convolutions. Therefore, it is non-linear, thus empowering the learning capability of deep models.
>
> In a word, our proposed “Fourier Up-Sampling” empowers the network not only more powerful non-linear learning capability but also more global (image-wide) dependency modelling capability.

---

### Official Review · Reviewer_mfGV · 2022-06-26

**Rating:** 8
**Confidence:** 5
**Soundness:** 4 excellent
**Presentation:** 4 excellent
**Contribution:** 4 excellent

**Summary:**

This paper tries to solve a challenging issue in frequency related neural network designs, i.e., how to model the multi-scale frequency patterns. Unlike the simplex up-sampling operations in the spatial domain, up-sampling frequency information in a neural network is non-trivial because the local property of spatial up-sampling cannot be directly used in the frequency domain. According to the spectral convolution theorem, this work proposes an interesting and theoretically sound solution, called Deep Fourier Up-Sampling (DFU). DFU is the first work to consider the modeling of multi-scale frequency patterns and provides key insights for the frequency related network designs. Besides, the DFU can be implemented by different variants with different theoretical support. The DFU is also flexible in its compatibility with existing neural networks. Extensive experiments on multiple vision tasks demonstrate the effectiveness of the proposed DFU.

**Questions:**

See the above weaknesses. The authors are suggested to give the detailed responses.

**Limitations:**

The authors adequately addressed the limitations and potential negative societal impact of their work.

**Strengths And Weaknesses:**

Strengths:
1. Interesting motivation: This work takes an interesting and challenging issue, multi-scale information modeling in the frequency domain, into account. Although there are some frequency-related neural networks, they cannot effectively and accurately model the relationship of multi-scale frequency information due to the Fourier transformation characteristics. However, the multi-scale frequency information is important for vision tasks. This key issue is the focus of this paper.
2. Theoretical proof: For the proposed Deep Fourier Up-Sampling, this work provides detailed and rigorous proof, making a theoretical sound solution. These proofs provide a new view to understand deep Fourier up-sampling and convince the readers.
3. Flexibility: The proposed Deep Fourier Up-sampling can be a generic operator, which can be compatibility with existing neural networks and boost their performance.
4. This paper conducts sufficient experiments to demonstrate the effectiveness of the proposed Deep Fourier Up-sampling. To show the versatility of the proposed method, the experimental comparisons are carried out on diverse vision tasks, including object detection, image segmentation, image de-raining, image de-hazing, and guided image super-resolution. The experiments on multiple vision tasks provide strong supports for the advantages of the proposed multi-scale frequency information modeling.


Weaknesses:
1. Although this paper provides sufficient experimental results, it would be better if the parameter comparisons and more visual results could be provided. The reviewer understands that this paper has conducted extensive experiments, covering low-level and high-level vision takes in the limited space. More results could be provided in the supplementary material.
2. In Section “Comparison and Analysis”, the underlying reasons why the FourierUp works better are insufficient. More explanations are expected. Furthermore, it is a little confusing that whether each variant of the FourierUp prefers a specified vision task? How to choose the variant when one uses the FourierUp for different vision tasks? Some insightful analysis or suggestion should be provided, which would guide the followers to make full use the capability of the proposed FourierUP.
3. It would be better if more visual results could be provided in the main paper rather than the supplementary material. It is fine to shrink the sizes of figures for showing more cases. Current format is inconvenient to switch between the main paper and the supplementary material for checking the visual results.
4. The flowchart of implementation details in supplementary materials are suggested to transfer to main manuscript.

---

> ### Author Response · Authors · 2022-08-01
> **Response to Reviewer mfGV**
>
> **1, The parameter comparisons and more visual results.**
>
> As suggested, we have added the complexity comparisons against the competitors, including trainable parameters and FLOPs. Due to the limited time, we only report some representative baselines: image de-raining LPNet and pan-sharpening DCFNET in the manuscript.
>
> |        |       | Original | Spatial-Up | FourierUp-AreaUp | FourierUp-Padding |
> |:----------------:|:----------------:|:----------------:|:----------------:|:----------------:|:-----------------:|
> |  LPNet | #Params |   27055  |    27127   |       27127      |       27127       |
> |        | FLOPs |  3.9621G |   3.9622G  |      3.9622G     |      3.9622G      |
> | DCFNET | #Params  |  2.7579M |   2.7988M  |      2.7988M     |      2.7988M      |
> |        | FLOPs | 13.7374G |  13.7376G  |     13.7376G     |      13.7376G     |
>
> Observing this table, we can find that our proposed “Fourier Up-Sampling” introduces negligible parameter and FLOPs. Therefore, the comparison further validates the effectiveness and potentials of our proposed general operator.
>
> In terms of the visual results, we will provide more results in the supplementary material.
>
>
>
> **2, The underlying reasons and whether each variant of the FourierUp prefers a specified vision task.**
>
> 1) the underlying reasons why the FourierUp works better are summarized as follows:
>
> First, our proposed “Fourier Up-Sampling” is performed in the frequency feature space where each factor connects with all the pixels in the spatial domain via Flourier transformation. In other words, learning in the frequency domain allows obtaining the image-wide receptive field that models the global contexts that are indispensable for vision tasks. Different from the existing spatial up-sampling strategies that only depend on local pixel attention for up-sampled reconstruction, our proposed “Fourier Up-Sampling” is performed in the frequency domain and thus fully explores the global dependency for up-sampling.
>
> Second, the spatial up-sampling is implemented by Transposed convolution and various interpolation techniques and both of them are linear operators. Therefore, the spatial up-sampling also can be regarded as a linear transformation. Differently, our “Fourier Up-Sampling” is implemented in the Fourier space and the transposed components are processed by different convolutions. Therefore, it is non-linear, thus empowering the learning capability of deep models.
>
> In a word, our proposed “Fourier Up-Sampling” empowers the network not only more powerful non-linear learning capability but also more global (image-wide) dependency modelling capability.
>
> 2) whether each variant of the FourierUp prefers a specified vision task?
>
> Like the spatial up-sampling techniques (e.g., transposed convolution, pixel shuffle, and various interpolation techniques), our proposed “Fourier Up-Sampling” is also the general strategy and each variant thus does not prefer a specified vision task. In addition, we have provided some insights on the use of different variants of the proposed “Fourier Up-Sampling” in Section 3.4 of the main paper.  As well recognized, existing global dependency techniques like non-local and transformer both require a huge computational cost. Instead, our proposed “Fourier Up-Sampling” is capable of modeling global relationships with simple implementation and cheap computational resources. Thus, one can try our operators in different tasks without considering the introduced computational burden.
>
> **3. The flowchart of implementation details.**
>
> Thank you for the suggestion. We will provide more results in the main paper and reorganize our layout. We will provide more details of our flowchart in the final version.

---

> ### Author Response · Authors · 2022-08-02
> **Response to Reviewer mfGV**
>
> We thank reviewer for the positive feedback. We are encouraged that the reviewer confirms the interesting motivation of our work, convincing theoretical proof, flexibility of the proposed operators, and sufficient experiments.

---

### Official Review · Reviewer_SXL9 · 2022-07-10

**Rating:** 5
**Confidence:** 5
**Soundness:** 3 good
**Presentation:** 3 good
**Contribution:** 3 good

**Summary:**

This paper proposes a theoretically feasible Deep Fourier Up-Sampling (FourierUp) to address up-sampling operators in Fourier domain. Extensive experiments across multiple computer vision tasks are conducted.

**Questions:**

Please see [Weaknesses].

**Limitations:**

Yes, the authors discussed the limitations in Sec. 5. However, I am also curious about the failure cases that the proposed method cannot handle currently.

**Strengths And Weaknesses:**

Strengths:
- Idea is interesting
- Easy to follow
- Clear writing

Weaknesses:
1. Sec. 3 introduces the details and theoretical evidences of the two kinds of FourierUp Modules, including Periodic Padding interpolation and Area Interpolation. However, it's not clear to me why the proposed FourierUp method is better. Even with the illustration in Fig. 1, the results of the two variants are not that good.  Thus, deeper analysis on why and how FourierUp helps improve the performance should be discussed and illustrated.

2. In experiments, more recent SOTA competitors are required to be compared. For example, PReNet is from CVPR'19, Aod-Net and PanNet are from ICCV'17. I think it's ok to adopt some classical methods as the baselines when proposing a novel framework for the existing task. It would be better if the proposed method is compared with the SOTA cutting-edge methods to provide some deeper understanding on the tasks, which may make the proposed method more insightful and convincing.

3. I am curious about what kind of failure cases the proposed method may have. Sec. 5 actually avoids to show such cases.

4. Complexity comparisons against the competitors are required.

---

> ### Author Response · Authors · 2022-08-02
> **Reponse to Reviewer SXL9 (part 3/3)**
>
> **3, Failures cases.**
>
> As a general operator, our method needs to be integrated with networks, such as the baseline networks used in our paper. We validate the effectiveness of our operator via comparing the performance gain with and without the operator. Thus, as a general operator, it is inapplicable to show failure cases as we are not sure whether the failure cases are caused by the original baselines or our operator. Instead, in terms of failure case, our method may have  limited application scenes. For example, our operator cannot be used in the networks that do not need the up-sampling operation, such as commonly used image classification networks, VGG, Resnet, and InceptionNet.
>
> **4, Complexity comparisons.**
>
> As suggested, we have added the complexity comparisons against the competitors, including trainable parameters and FLOPs. Due to the limited time, we only report some representative baselines：image de-raining LPNet and pan-sharpening DCFNET in the manuscript.
>
> |        |       | Original | Spatial-Up | FourierUp-AreaUp | FourierUp-Padding |
> |:------:|:-----:|:--------:|:----------:|:----------------:|:-----------------:|
> |  LPNet | #Params |   27055  |    27127   |       27127      |       27127       |
> |        | FLOPs |  3.9621G |   3.9622G  |      3.9622G     |      3.9622G      |
> | DCFNET | #Params |  2.7579M |   2.7988M  |      2.7988M     |      2.7988M      |
> |        | FLOPs | 13.7374G |  13.7376G  |     13.7376G     |      13.7376G     |
>
> Observing this table, we can find that our proposed “Fourier Up-Sampling” introduces negligible parameter and FLOPs increment and achieves the significant performance gain as suggested in the experimental parts of the main manuscript when they are compared with the original models. Therefore, the comparison further validates the effectiveness and potentials of our proposed general operator.

---

> ### Author Response · Authors · 2022-08-02
> **Reponse to Reviewer SXL9 (part 2/3)**
>
> **2,  More recent SOTA competitors.**
>
> As suggested, we  have added the state-of-the-art cutting-edge methods as the baselines to validate the effectiveness of our proposed “Fourier Up-Sampling”. Due to limited time, we only select the latest image de-raining method SGCN [1] to conduct experimental validation on Rain200L and Rain200H datasets. As can be seen in this table, with the proposed Fourier Up-sampling operators, the performance of original SGCN is improved in terms of PSNR and SSIM. The results are consistent with other vision tasks presented in our main paper. We will add more baselines to further show the effectiveness of our method in the final version.
>
> |        |                   | Rain200L |       | Rain200H |       |
> |:------:|:-----------------:|:--------:|:-----:|:--------:|:-----:|
> | method |   configuration   |   PSNR   |  SSIM |   PSNR   |  SSIM |
> |        |      Original     |   37.65  | 0.976 |   29.13  | 0.895 |
> |  SGCN  |     Spatial-Up    |   37.86  | 0.982 |   29.47  | 0.903 |
> |        |  FourierUp-AreaUp |   39.57  | 0.985 |   29.82  | 0.908 |
> |        | FourierUp-Padding |   39.83  | 0.987 |   29.95  | 0.913 |
>
> [1] Xueyang Fu, et.al. Successive Graph Convolutional Network for Image De-raining. IJCV, 2022.

---

> ### Author Response · Authors · 2022-08-02
> **Reponse to Reviewer SXL9 (part 1/3)**
>
> **1, Figure 1.**
>
> First, Figure-1 aims to illustrate the motivation of our work.  Unlike the spatial domain with local similarity property, up-sampling in the Fourier domain is more challenging as it does not follow such a local similarity property. Specifically, in Figure-1 (a), the input image is passed to the down-sampling and up-sampling in the spatial domain, and it still maintains the structure of the original input image. In other word, the resulting image can be added to the original input image because of the coordinate-aware pixel alignment. However, as shown in Figure-1 (b), directly employing the spatial up-sampling to process the down-sampled frequency information cannot restore the coordinate-aware structure alignment, thus incapable of integrating it with the original input image for multi-scale frequency modeling. Inspired by this issue, we attempt to design a more ingenious “Fourier Up-Sampling”. In Figure-1 (c), it can be seen that the proposed “Fourier Up-Sampling” can effectively restore the image of coordinate-aware structure alignment with the original input image.
>
> Second, in terms of the question “reconstructions in the spatial domain are really bad after periodic padding or area-interpolation. Why does it still work?”, the reviewer may misunderstand our Figure-1. Figure-1 just aims to illustrate the motivation of our paper, but does not completely correspond to our detailed designs.
>
> To be specific, (1) the reconstruction of Figure-1 (c) in the spatial domain is incompletely. It only employs the core step of periodic padding or area-interpolation of our proposed “Fourier Up-Sampling” for illustration and has not employed the following transformation coefficients with 4 and $\frac{-4\pi}{N}sin(\frac{\pi y}{N})(1+cos(\frac{\pi x}{M}))$ that are inferred in Theorem-1 and Theorem-2 of the main paper. When adding this step. i.e., complete reconstruction, our proposed method can produce visually pleasing reconstruction.
>
> (2) For visualization, Figure-1 shows the results in the image space and only adopts the fixed transformations. However, our proposed “Fourier Up-Sampling” is targeted at the feature space. In the feature space, we formulate our method as the learnable transformations for better flexibility and accuracy. Note that albeit being designed on the basis of strict theories, both constructed frequency up-sampling modules contain certain approximations, such as a learnable convolution operator instead of strictly 1/4 as described in Theorem-1 of the main manuscript, and an approximation cropping to preserve the map corners instead of accurate $\frac{-4\pi}{N}sin(\frac{\pi y}{N})(1+cos(\frac{\pi x}{M}))$ mapping as proved in Theorem-2 of the main manuscript and Theorem-3 of the supplementary materials. Such approximations make the proposed modules feasible and flexible when they are used to represent the frequency structures of real data. It is worth noting that this is the first attempt for constructing easy equitable frequency up-sampling modules. With our exploration, we wish this work could  arouse more effective and rational designs for frequency up-sampling.
>
> (3) Compared with spatial up-sampling, our proposed “Fourier Up-Sampling” has the following advantages. In the feature space, the spatial up-sampling only depends on local pixel attention and can be regarded as a linear transformation. Instead, our proposed “Fourier Up-Sampling” is performed in the frequency domain and thus fully explores the global contexts via Fourier transformation, which achieves better up-sampling than simple linear transformation of spatial up-sampling. Furthermore, compared with the spatial up-sampling, our “Fourier Up-Sampling” is non-linear, thus empowering the learning capability of a deep model.
>
> In summary, Figure-1 aims to illustrate the motivation of our paper not completely corresponds to our designed “Fourier Up-Sampling” solutions.

---

> ### Author Response · Authors · 2022-08-02
> **Reponse to Reviewer SXL9**
>
> We thank reviewer for the positive feedback. We are encouraged that the reviewer confirms the interesting idea of our work, easy to follow, and clear writing.

---

### Official Review · Reviewer_XGiH · 2022-07-11

**Rating:** 8
**Confidence:** 5
**Soundness:** 4 excellent
**Presentation:** 4 excellent
**Contribution:** 4 excellent

**Summary:**

This paper proposes a novel up-scaling strategy to upscale the spatial features in an encoder-decoder-based spatial up-down sampling network. Upscaling is performed on the frequency domain to explore the global dependency. The upscaling module has been experimented with several computer vision tasks and experimental results show significant performance improvement in those tasks.

**Questions:**

Why did this paper not explore the frequency down-sampling strategies and experiments with different variants? Did frequency down-sampling not give satisfactory results? If so, why does only frequency upscaling work?

**Ethics Review Area:**

["I don’t know"]

**Strengths And Weaknesses:**

This paper mainly proposes two different types of Fourier upscaling, namely periodic padding and area-interpolation. The main strengths are as follows.
1. This paper explored the relationship between spatial and frequency domains. The theoretical evidence in support of upscaling module design is explained properly.
2. The experiments in comparison with existing upscaling and different variants of Fourier upscaling are performed. The experimental outcome shows promising performance.
3. The paper writing and explanations are good. I actually had this kind of idea but never tried this. It is refreshing to see that this paper has done this. This work will drive the community in a certain direction.

There is no such major weakness in this paper. I would like to know an explanation behind those results as we can see from Figure 1 that reconstructions in the spatial domain are really bad after periodic padding or area-interpolation. Why does it still work?

---

> ### Author Response · Authors · 2022-08-01
> **Reponse to Reviewer XGiH**
>
> **1, Figure 1.**
>
> First, Figure-1 aims to illustrate the motivation of our work.  Unlike the spatial domain with local similarity property, up-sampling in the Fourier domain is more challenging as it does not follow such a local similarity property. Specifically, in Figure-1 (a), the input image is passed to the down-sampling and up-sampling in the spatial domain, and it still maintains the structure of the original input image. In other word, the resulting image can be added to the original input image because of the coordinate-aware pixel alignment. However, as shown in Figure-1 (b), directly employing the spatial up-sampling to process the down-sampled frequency information cannot restore the coordinate-aware structure alignment, thus incapable of integrating it with the original input image for multi-scale frequency modeling. Inspired by this issue, we attempt to design a more ingenious “Fourier Up-Sampling”. In Figure-1 (c), it can be seen that the proposed “Fourier Up-Sampling” can effectively restore the image of coordinate-aware structure alignment with the original input image.
>
> Second, in terms of the question “reconstructions in the spatial domain are really bad after periodic padding or area-interpolation. Why does it still work?”, the reviewer may misunderstand our Figure-1. Figure-1 just aims to illustrate the motivation of our paper, but does not completely correspond to our detailed designs.
>
> To be specific, (1) the reconstruction of Figure-1 (c) in the spatial domain is incompletely. It only employs the core step of periodic padding or area-interpolation of our proposed “Fourier Up-Sampling” for illustration and has not employed the following transformation coefficients with 4 and $\frac{-4\pi}{N}sin(\frac{\pi y}{N})(1+cos(\frac{\pi x}{M}))$ that are inferred in Theorem-1 and Theorem-2 of the main paper. When adding this step. i.e., complete reconstruction, our proposed method can produce visually pleasing reconstruction.
>
> (2) For visualization, Figure-1 shows the results in the image space and only adopts the fixed transformations. However, our proposed “Fourier Up-Sampling” is targeted at the feature space. In the feature space, we formulate our method as the learnable transformations for better flexibility and accuracy. Note that albeit being designed on the basis of strict theories, both constructed frequency up-sampling modules contain certain approximations, such as a learnable convolution operator instead of strictly 1/4 as described in Theorem-1 of the main manuscript, and an approximation cropping to preserve the map corners instead of accurate $\frac{-4\pi}{N}sin(\frac{\pi y}{N})(1+cos(\frac{\pi x}{M}))$ mapping as proved in Theorem-2 of the main manuscript and Theorem-3 of the supplementary materials. Such approximations make the proposed modules feasible and flexible when they are used to represent the frequency structures of real data. It is worth noting that this is the first attempt for constructing easy equitable frequency up-sampling modules. With our exploration, we wish this work could  arouse more effective and rational designs for frequency up-sampling.
>
> (3) Compared with spatial up-sampling, our proposed “Fourier Up-Sampling” has the following advantages. In the feature space, the spatial up-sampling only depends on local pixel attention and can be regarded as a linear transformation. Instead, our proposed “Fourier Up-Sampling” is performed in the frequency domain and thus fully explores the global contexts via Fourier transformation, which achieves better up-sampling than simple linear transformation of spatial up-sampling. Furthermore, compared with the spatial up-sampling, our “Fourier Up-Sampling” is non-linear, thus empowering the learning capability of a deep model.
>
> In summary, Figure-1 aims to illustrate the motivation of our paper not completely corresponds to our designed “Fourier Up-Sampling” solutions.
>
>
> **2，About frequency down-sampling.**
>
> Thanks for your professional comments. As we illustrated in the manuscript, both the frequency down-sampling and up-sampling strategies contribute to multi-scale frequency modeling. Frequency down-sampling is also important. However, the frequency down-sampling has been studied in previous work [1] while the frequency up-sampling has never been explored. To this end, we are encouraged to fill this gap. The core of this work is frequency up-sampling.
>
> Furthermore, the core derivations between the down-sampling in [1] and our proposed up-sampling are significantly different. Specifically, the frequency down-sampling proposed in [1] is heuristic and only implemented by cropping the center-around frequency information as the expected down-sampled size. In contrast, our frequency up-sampling is on the basis of strict theories.
>
> [1] Oren Rippel, Jasper Snoek, and Ryan P. Adams. 2015. Spectral representations for convolutional neural networks. Neural Information Processing Systems (NIPS'15).

---

> ### Author Response · Authors · 2022-08-02
> **Reponse to Reviewer XGiH**
>
> We thank the reviewer for the positive feedback. We are encouraged that the reviewer confirms the theoretical evidence provided in our work, the promising performance of our method, the good writing and explanations of our paper. We also happy to see the reviewer had this kind of idea before.

---

### Meta-Review · Area_Chair_2qdn · 2022-08-24

**Recommendation:** Accept
**Confidence:** Certain

**Metareview:**

This paper proposes using Fourier up-sampling for multi-scale modeling. The paper received initial scores of 8 8 5 3. After the rebuttal and in-depth discussions, most reviewers are satisfied with the authors' replies. Reviewer-tPpX who gives negative scores still has concerns about the novelty and presentation of this paper. As I discussed with the reviewer, we finally did not find similar works that share the similar idea as this paper, so I recommend acceptance for this paper. However, I agree with the reviewer-tPpX that the presentation of this paper should be improved. There are actually too many equations and derivations in the manuscript, and this is not friendly for a NeurIPS paper. Please remember to move them into the appendix as much as possible if preparing for the camera-ready version.

**Award:**

No

---

### Decision · Program_Chairs · 2022-09-14

Accept